# BEYOND FAST AND SLOW: COGNITIVE-INSPIRED ELASTIC REASONING FOR LARGE LANGUAGE MODELS

## ABSTRACT

Large language models (LLMs) have demonstrated impressive performance across various language tasks. However, existing LLM reasoning strategies mainly rely on the LLM itself with fast or slow mode (like o1 thinking) and thus struggle to balance reasoning efficiency and accuracy across queries of varying difficulties. In this paper, we propose **Cog**nitive-Inspired **E**lastic **R**easoning (**CogER**), a framework inspired by human hierarchical reasoning that dynamically selects the most suitable reasoning strategy for each query. Specifically, CogER first assesses the complexity of incoming queries and assigns them to one of several predefined levels, each corresponding to a tailored processing strategy, thereby addressing the challenge of unobservable query difficulty. To achieve automatic strategy selection, we model the process as a Markov Decision Process and train a CogER-Agent using reinforcement learning. The agent is guided by a reward function that balances solution quality and computational cost, ensuring resource-efficient reasoning. Moreover, for queries requiring external tools, we introduce Cognitive Tool-Assisted Reasoning, which enables the LLM to autonomously invoke external tools within its chain-of-thought. Extensive experiments demonstrate that CogER outperforms state-of-the-art Test-Time scaling methods, achieving at least a 13% relative improvement in average exact match on In-Domain tasks and an 8% relative gain on Out-of-Domain tasks.

## 1 INTRODUCTION

Large language models (LLMs), such as ChatGPT (Achiam et al., 2023) and DeepSeek (Guo et al., 2025), have achieved impressive results on many tasks, including multi-turn dialogue (Stark et al., 2023) and embodied intelligence (Mu et al., 2023). However, as model size and the number of inference tokens increase, the computational resources required for inference grow substantially, creating a major bottleneck for real-world applications. Meanwhile, user queries vary widely in complexity, from straightforward fact-based questions to multi-hop reasoning tasks, and in some cases, even require external tool invocation. This diversity makes traditional LLM reasoning approaches, rooted in the dual-process theory of fast (System 1) and slow (System 2) thinking, face critical limitations in handling all types of queries efficiently and effectively (Li et al., 2025). Consequently, it is crucial to dynamically allocate reasoning strategies based on query complexity in practical applications.

*Unfortunately*, existing LLMs typically apply a uniform reasoning process regardless of query complexity (Aggarwal & Welleck, 2025). This one-size-fits-all reasoning strategy risks either wasting computation on trivial inputs or inadequately handling more demanding queries. Achieving flexible and efficient reasoning requires addressing two key challenges: 1) *Unforeseen query difficulty*: The true complexity of an incoming query is often not observable in advance, making it difficult to allocate computational resources dynamically and appropriately. 2) *Cost–quality trade-off*: Larger language models generally yield higher accuracy but incur substantially greater compute costs, forcing a careful balance between performance and efficiency along the Pareto frontier.

Recently, several attempts (Jiang et al., 2023; Dong et al., 2024; Du et al., 2023; Ong et al., 2025a; Yang et al., 2025b) have been proposed to tailor reasoning strategies to downstream task demands, which can be broadly divided into the following categories: 1) *LLM ensemble methods* (Jiang et al., 2023; Dong et al., 2024; Du et al., 2023) often combine outputs from multiple candidate models to boost accuracy. However, each input must typically be processed by all models in the ensemble,

leading to substantial computational overhead. 2) *Test-time scaling methods* (Muennighoff et al., 2025; Yang et al., 2025b; Snell et al., 2024; Aggarwal & Welleck, 2025)adapt reasoning costs based on the estimated difficulty of inputs, for instance by adjusting the length of chain-of-thought (CoT) reasoning or employing early-exit mechanisms. While more efficient, these methods often struggle to assess difficulty accurately for all queries and lack adaptive mechanisms for invoking external tools. As a result, they fall short in handling complex tasks requiring access to additional knowledge sources, limiting their flexibility and extensibility in real-world applications.

To address these limitations, we propose the **Cog**nitive-Inspired **E**lastic **R**easoning (**CogER**) framework for efficient scaling of language model reasoning. This framework dynamically selects the most suitable processing mode for each query based on its complexity. Specifically, inspired by **Bloom's Taxonomy** (Bloom et al., 1956), we first categorize incoming queries into four complexity levels ($L_1 - L_4$), each associated with a tailored reasoning strategy, thereby mitigating the challenge of unforeseen query difficulty. Then, we model the strategy selection process as a Markov Decision Process (MDP), in which a CogER-Agent chooses one of four actions (No Think, Think, Extend, or Delegate) to process each query, based on the predicted complexity level. To guide the training of this agent, we design a reward function that explicitly balances computational cost against output quality, ensuring that each query receives only the computational resources commensurate with its complexity. Finally, for $L_4$ queries that require external knowledge, we introduce Cognitive Tool-Assisted Reasoning (CoTool), enabling the LLM to autonomously invoke external tools at appropriate points within its chain-of-thought, enabling flexible and knowledge-augmented reasoning.

**Main novelty and contributions. 1)** We propose **Cog**nitive-Inspired **E**lastic **R**easoning (**CogER**), which dynamically selects the most appropriate processing mode for each query. It classifies incoming queries into four complexity levels, formulates reasoning strategy selection as an MDP, and introduces a novel reward function to train a CogER-Agent that dynamically selects the optimal strategy under constrained computational budgets. **2)** We introduce **CoTool**, which enables the model to autonomously decide when and how to invoke external tools during complex reasoning, seamlessly integrating API calls within its CoT, and we provide the **RSTKit toolkit** to facilitate this process. **3)** Extensive experiments demonstrate that, compared to SOTA TTS methods, CogER achieves at least a 13% relative improvement in average $EM$ on ID tasks and an 8% relative gain on OOD tasks.

## 2 RELATED WORK

**Large language models (LLMs) ensemble methods** (Chen et al., 2025a) aim to combine multiple models to leverage their complementary strengths. Existing approaches can be categorized into three paradigms based on integration timing: ensemble-before-inference, ensemble-during-inference, and ensemble-after-inference. Ensemble-before-inference methods (Lu et al., 2024a; Ding et al., 2024; Srivatsa et al., 2024; Lu et al., 2024b) first apply a routing mechanism, either pretrained on custom data or trained on the fly, to dispatch each query to the most suitable, specialized model, thereby enabling more cost-efficient inference. Ensemble-during-inference methods (Huang et al., 2024; Xu et al., 2025b; Park et al., 2025) combine outputs from multiple models at different levels of granularity, including the token level, span level, and reasoning-step level, and then merge the resulting text segments back into the decoding context to iteratively refine the output. Ensemble-after-inference methods (Park et al., 2025; Hu et al., 2025; Du et al., 2023) generate complete responses independently from each candidate LLM and then consolidate them via ranking, majority voting, or fitness scoring to select the highest-quality output for final delivery. In contrast, our CogER learns a lightweight policy network to dispatch each query to a single, optimal inference action, No_Think, Think, Extend, Delegate, within an MDP, thereby enabling fine-grained per-query adaptation.

**Test-Time Scaling (TTS) methods** optimize computational resource allocation during inference through adaptive reasoning depth control. Muennighoff et al. (2025) propose budget forcing, a technique to regulate the computation of test time by prematurely stopping the CoT of the model or extending it by repeated insertion of the token 'wait' when the model attempts to terminate generation. Aggarwal & Welleck (2025) propose LCPO, a straightforward reinforcement learning (RL) approach designed to maximize accuracy while respecting user-specified length constraints. Yang et al. (2025b) propose Thinking-Optimal Scaling, which trains the model on seed examples with varied response lengths to learn appropriate reasoning efforts and then self-improves by selecting the shortest correct responses on new tasks. In contrast to TTS methods that only adjust reason-

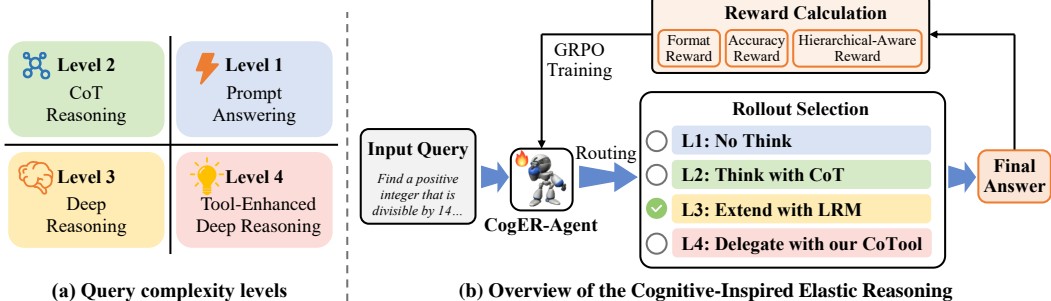

Figure 1: (a) Query complexity levels. (b) Overview of the CogER. Given an input query, the CogER-Agent selects a complexity level ($L_1 - L_4$) and routes it to the corresponding reasoning strategy, including direct answering, light to multi-step reasoning, and Cognitive Tool-Assisted Reasoning. The CogER-Agent is trained via GRPO with a composite reward that combines Format Reward $\mathcal{R}_{\text{format}}$, Accuracy Reward $\mathcal{R}_{\text{accuracy}}$, and Hierarchical-Aware Reward $\mathcal{R}_{\text{hierarchy}}$.

ing depth based on coarse difficulty estimates, CogER selects among diverse reasoning modes and seamlessly integrates external tool usage, achieving more versatile resource allocation.

## 3 PROBLEM STATEMENT AND MOTIVATION

Given a set of user queries $X = \{x_1, \ldots, x_K\}$, we seek to process each query $x_i$ with a reasoning strategy that minimizes computational cost while maximizing solution quality. Specifically, for each $x_i$, we select reasoning actions $a_i \in \mathcal{A} = \{\text{No\_Think}, \text{Think}, \text{Extend}, \text{Delegate}\}$, where No_Think uses a lightweight LLM to produce an immediate answer; Think invokes internal multi-step reasoning within a moderately sized LLM; Extend performs test-time scaling by engaging a Large Reasoning Model (LRM) to generate a longer chain of thought; Delegate invokes parameterized external tools (*e.g.*, search engines, calculator) to obtain intermediate information, which is seamlessly incorporated into the model's reasoning process to produce the final output. Each action $a \in \mathcal{A}$ incurs a computational cost $C(a)$ and achieves an expected solution quality $\alpha(a)$. Our objective is to learn a policy $\pi$ mapping each query $x_i$ to exactly one reasoning action to minimize the combined cost–accuracy loss:

$$\min_{\pi} \sum_{i=1}^{K} \Big[ C\big(\pi(x_i)\big) - \big(\pi(x_i)\big) \Big]. \tag{1}$$

**Motivation**. In real-world applications, user questions exhibit a wide range of complexity. For example, some queries can be answered in a single step, whereas others require deep, multi-step reasoning or integration of external information sources. However, existing LLMs apply the same reasoning procedure to every query with a high computational cost, which may lead to *wasted resources on simple queries and poor performance on complex ones* (Sui et al., 2025; Hu et al., 2025).

To address this issue, we provide per-query adaptivity by selecting the most appropriate processing mode for each question. Simple lookups invoke a lightweight model to generate an immediate answer. Moderately difficult queries trigger internal reasoning within a medium-sized model. Harder queries use a large reasoning model to produce an extended chain of thought. Finally, queries beyond the model's standalone capabilities call external tools or APIs and incorporate their outputs. This dynamic selection mechanism reduces overall cost while preserving or improving accuracy, enabling efficient and scalable reasoning across diverse workloads.

## 4 COGNITIVE-INSPIRED ELASTIC REASONING FOR LLMS

In this paper, we propose the **Cog**nitive-Inspired **E**lastic **R**easoning (**CogER**) for efficient scaling of language model reasoning, which dynamically selects the most appropriate processing mode for each query. The overall framework is in Figure 1. Given an input query, the CogER-Agent selects a complexity level ($L_1 - L_4$) and routes it to the corresponding reasoning strategy, including direct answering, light to multi-step reasoning, and Cognitive Tool-Assisted Reasoning (*c.f.* Sec. 4.4).

### 4.1 QUERY COMPLEXITY CLASSIFICATION

To efficiently allocate reasoning strategies based on the diverse computational requirements of different queries, we draw inspiration from **Bloom's Taxonomy** (Bloom et al., 1956) to classify queries by their cognitive demand. Specifically, we define four levels of query complexity, denoted as $L_1$, $L_2$, $L_3$, and $L_4$, with each level representing an increasing degree of reasoning depth and computational demand, as illustrated in Figure 1(a). The details of each level are as follows:

- $L_1$**: Prompt Answering.** Queries with simple, unambiguous structure that require no reasoning and can be answered directly (*e.g.*, "$2 + 2 = ?$"). [*Corresponds to Bloom's "Remember" level.*]
- $L_2$**: CoT Reasoning.** Queries that demand basic comprehension and simple reasoning (*e.g.*, "How many minutes are in 3.5 hours?"). [*Corresponds to Bloom's "Understand/Apply" levels.*]
- $L_3$**: Deep Reasoning.** Queries requiring multi-hop reasoning, analysis, or evidence weighing (*e.g.*, "Analyze the trends in data table"). [*Corresponds to Bloom's "Analyze/Evaluate" levels.*]
- $L_4$**: Tool-Enhanced Deep Reasoning.** Queries that require creative synthesis of information to generate novel solutions (*e.g.*, "Formulate a proof strategy for the Collatz conjecture."). [*Corresponds to Bloom's "Create" level.*]

This classification facilitates a principled allocation of computational resources: lower-complexity queries (*e.g.,* $L_1$ and $L_2$) can be handled by lightweight reasoning modules, while higher-complexity queries (*e.g.,* $L_3$ and $L_4$) may demand more sophisticated reasoning techniques or assistance from external tools. By tailoring the reasoning strategy to the cognitive complexity of each query, the system can achieve more efficient use of computational resources and improved overall performance.

### 4.2 COGNITIVE-INSPIRED ELASTIC REASONING AS MARKOV DECISION PROCESS

We seek to design a CogER-Agent that dynamically selects reasoning strategies based on the complexity of each query, optimizing the balance between computational cost and solution quality. Dynamic reasoning over diverse queries naturally constitutes a sequential decision-making problem under uncertainty: the agent must choose among multiple reasoning operations step by step to balance resource expenditure with answer accuracy. Such a process aligns perfectly with the Markov Decision Process (MDP) (Van Otterlo & Wiering, 2012), which seeks a policy that maximizes expected cumulative utility. Therefore, we model it as a MDP: $< \mathcal{S}, \mathcal{A}, \mathcal{T}, \mathcal{R}, \pi >$. The state space of the environment is $\mathcal{S}$ and the action space of the agent is $\mathcal{A}$. At time step $t$, the agent takes the state $s_t \in \mathcal{S}$ as input and performs an action $a_t \in \mathcal{A}$ through the policy network $\pi : \mathcal{S} \times \mathcal{A} \to [0, 1]$. The environment changes to the next state $s_{t+1} = \mathcal{T}(s_t, a_t)$ according to the transition function $\mathcal{T}$ and a reward $r_t = \mathcal{R}(s_t, a_t)$ is received with reward function $\mathcal{R}$. The MDP is as follows:

**States** $\mathcal{S}$ is a set of states which describe the environment. At time step $t$, the state can be represented as $s_t = [x, y_{1:t-1}, L_i]$, where $x$ denotes the input query, $y_{1:t-1}$ represents the natural language output at time steps 1 through $t - 1$, and $L_i \in L = \{L_1, L_2, L_3, L_4\}$ denotes the inferred task complexity level corresponding to the query. Note that the complexity level $L_i$ may not be presented at every time step $t$, as the model may infer this level based on the context or internal reasoning.

**Actions** $\mathcal{A}$ is a set of actions that the agent can take to process the query. Each action corresponds to a different reasoning strategy based on the complexity of the query. The action space includes both the vocabulary space, from which the model generates tokens, and predefined reasoning strategies for different complexity levels. Specifically, the action space consists of: $\mathcal{A} = \{\text{No\_Think}, \text{Think}, \text{Extend}, \text{Delegate}, \mathcal{V}\}$, where $\mathcal{V}$ represents the vocabulary of possible words or tokens the model can generate as part of its reasoning, and the other actions are strategies the agent can apply based on the query's complexity.

**Rewards** $\mathcal{R}(\mathcal{S}, \mathcal{A})$ is the reward function. In this setting, the reward can be considered as a composite signal that rewards correctly formatted strategy outputs and incentivizes high accuracy with minimal resource consumption. The details of the reward function are given in the Sec. 4.3.

**Policy** $\pi_\theta(a|s) : \mathcal{A} \times \mathcal{S} \to [0, 1]$ describes the behaviors of the agent. The agent takes the current state $s_t$ as input and outputs a probability distribution for each possible action $a_t \in \mathcal{A}$:

$$\pi(a_t = i|s_t; \theta) = \frac{\exp\left(f_\theta(s_t)_i\right)}{\sum_{j=1}^{N} \exp\left(f_\theta(s_t)_j\right)}, \tag{2}$$

where $f_\theta(s_t)$ is the output vector of the policy with input $s_t$, and $i$ denotes the index of the action.

**CogER Rollout.** To enable the agent to generate reasoning trajectories and select appropriate reasoning strategies autonomously, we adopt a dedicated system prompt (*c.f.* App. B.1) to guide the thinking of the model during rollout. This prompt instructs the model to wrap each incoming query with special tokens, such as `<question_level>` and `</question_level>` to explicitly mark its complexity. In implementation, this leads to a two-stage decision process: the CogER-Agent first selects one of the four high-level reasoning modes $L_1$–$L_4$ by emitting a task-level tag (e.g., `<question_level>` $L_2$ `</question_level>`), and then, conditioned on this choice, the underlying LLM generates the full response autoregressively over the vocabulary $\mathcal{V}$ within the selected mode. Once the level is identified, the agent proceeds as follows:

- $L_1$**-level**: No_Think. Return the answer immediately with no reasoning.
- $L_2$**-level**: Think. Apply a chain-of-thought strategy (Wei et al., 2022b) using a moderately sized LLM to produce a concise reasoning trail.
- $L_3$**-level**: Extend. Produce an extended chain-of-thought with a large reasoning model.
- $L_4$**-level**: Delegate. Invoke external tools via our CoTool (*c.f.* Sec. 4.4) to support reasoning.

By dynamically adjusting its strategy according to the query complexity, the CogER-Agent can achieve a better trade-off between computation overhead and reasoning performance.

### 4.3 REWARD FUNCTION DESIGN

In our MDP, we define the reward as a composite of three components: Format Reward, Accuracy Reward, and Hierarchical-Aware Reward. These components encourage the agent to generate formatted level tags correctly, achieve high answer accuracy, and avoid unnecessary use of overly complex strategies. Formally, the reward $\mathcal{R}(\mathcal{S}, \mathcal{A})$ is defined as follows:

$$\mathcal{R}(\mathcal{S}, \mathcal{A}) = \mathcal{R}_{\text{format}}(\mathcal{S}, \mathcal{A}) + \mathcal{R}_{\text{accuracy}}(\mathcal{S}, \mathcal{A}) + \mathcal{R}_{\text{hierarchy}}(\mathcal{S}, \mathcal{A}), \quad (3)$$

where $\mathcal{R}_{\text{format}}(\cdot)$ is Format Reward, $\mathcal{R}_{\text{accuracy}}(\cdot)$ is Accuracy Reward, and $\mathcal{R}_{\text{hierarchy}}(\cdot)$ is Hierarchical-Aware Reward. In practice, if the output does not satisfy the required format and we cannot reliably extract a valid task-level tag, the trajectory is treated as invalid for level-specific evaluation, and both $\mathcal{R}_{\text{accuracy}}(\cdot)$ and $\mathcal{R}_{\text{hierarchy}}(\cdot)$ are set to 0.

**Format Reward $\mathcal{R}_{\text{format}}$.** The Format Reward encourages the agent to generate outputs with the correct structural format, specifically ensuring the inclusion of a properly placed task-level tag (*i.e.,* `<question_level>`$L_i$`</question_level>`) that corresponds to the query's complexity level, which can be formulated as follows:

$$\mathcal{R}_{\text{format}}(\mathcal{S}, \mathcal{A}) = \begin{cases} +1, & \text{if all required fields appear and are in the correct order} \\ 0, & \text{otherwise} \end{cases}. \quad (4)$$

**Accuracy Reward $\mathcal{R}_{\text{accuracy}}$.** The Accuracy Reward encourages the agent to produce correct answers by assigning a positive reward only when the predicted result matches the expected outcome:

$$\mathcal{R}_{\text{accuracy}}(\mathcal{S}, \mathcal{A}) = \begin{cases} +1, & \text{if the final answer is correct} \\ 0, & \text{otherwise} \end{cases}. \quad (5)$$

**Hierarchical-Aware Reward $\mathcal{R}_{\text{hierarchy}}$.** The Hierarchical-Aware Reward encourages the agent to solve queries with the simplest sufficient strategy, thereby avoiding unnecessary computational overhead. Specifically, the reward assigns a base credit for using each reasoning level and penalizes the use of unnecessarily complex strategies when a simpler level suffices. The reward is defined as:

$$\mathcal{R}_{\text{hierarchy}}(\mathcal{S}, \mathcal{A}) = b(L_{\min}(\mathcal{S})) - \delta(L_{\min}(\mathcal{S}), L(\mathcal{S})), \quad (6)$$

where $L(\mathcal{S})$ denotes the selected reasoning level, and $L_{\min}(\mathcal{S})$ is the minimal level required to solve the given query. The base credit $b(L(\mathcal{S}))$ increases linearly with the reasoning level:

$$b(L_{\min}(\mathcal{S})) = 0.5 \cdot (L_{\min}(\mathcal{S}) - 1), \quad L_{\min}(\mathcal{S}) \in \{1, 2, 3, 4\}. \quad (7)$$

The penalty term is defined as:

$$\delta(L_{\min}(\mathcal{S}), L(\mathcal{S})) = 0.2 \cdot (L(\mathcal{S}) - L(\mathcal{S})_{\min})_+, \quad (8)$$

---

**Algorithm 1** The pipeline of Cognitive Tool-Assisted Reasoning

---

**Input:** Reasoning Model $\mathcal{M}$, Questions $Q$, Task instruction $I$, Reason-in-tool instruction $I_{\text{tool}}$.

1: Initialize set of unfinished sequences $\mathcal{S} \leftarrow \{I \oplus q \mid q \in Q\}$, set of finished sequences $\mathcal{F} \leftarrow \{\}$
2: **while** $\mathcal{S} \neq \emptyset$ **do**
3:     Generate all sequences in $\mathcal{S}$ until EOS or `<|end_tool_query|>`: $\mathcal{T} \leftarrow \mathcal{M}(\mathcal{S})$
4:     Initialize empty set $\mathcal{S}_r \leftarrow \{\}$
5:     **for** each sequence Seq $\in \mathcal{T}$ **do**
6:         **if** Seq ends with `<|end_tool_query|>` **then**
7:             Extract tool query: $q_{\text{tool}} \leftarrow$ Extract(Seq, `<|begin_tool_query|>`, `<|end_tool_query|>`)
8:             Retrieve tool execution results: $T_{\text{results}} \leftarrow$ `SearchAndExecuteTools`$(q_{\text{tool}})$
9:             Construct input for Reason-in-tools: $I_T \leftarrow I_{\text{tool}} \oplus q_{\text{tool}} \oplus$ Seq $\oplus T_{\text{results}}$
10:             **Append** the tuple $(I_T, \text{Seq})$ to $\mathcal{S}_r$
11:         **else if** Seq ends with EOS **then**
12:             Remove Seq from $\mathcal{S}$, add Seq to $\mathcal{F}$
13:     **if** $\mathcal{S}_r \neq \emptyset$ **then**
14:         Prepare batch inputs: $\mathcal{I}_r \leftarrow \{I_T \mid (I_T, \text{Seq}) \in \mathcal{S}_r\}$
15:         Reason-in-Tool: $\mathcal{T}_r \leftarrow \mathcal{M}(\mathcal{I}_r)$
16:         **for** $i \leftarrow \{1, \ldots, |\mathcal{T}_r|\}$ **do**
17:             Let $r \leftarrow \mathcal{T}_r[i]$, Seq $\leftarrow \mathcal{S}_r[i].$Seq
18:             Let $r_{\text{final}} \leftarrow$ `<|begin_tool_result|>` $\oplus r \oplus$ `<|end_tool_result|>`
19:             Update sequence Seq in $\mathcal{S}$: Seq $\leftarrow$ Seq $\oplus r_{\text{final}}$

**Output:** Finished Sequences $\mathcal{F}$

---

where $(\cdot)_+ = \max(\cdot, 0)$ ensures that penalties are only applied when the selected level exceeds the minimal sufficient one. This design encourages correct answers with minimal reasoning cost while discouraging the overuse of higher-level strategies. As an example, consider a query that can be solved at all levels $\{L_1, L_2, L_3, L_4\}$. The resulting rewards $\mathcal{R}_{\text{format}}(\mathcal{S}, \mathcal{A})$ are $\{L_1 = 0, L_2 = -0.2, L_3 = -0.4, L_4 - 0.6\}$. This shows that the reward favors the minimal sufficient level while penalizing unnecessary complexity.

## 4.4 Cognitive Tool-Assisted Reasoning

To address complex problems that require up-to-date knowledge, precise computation, or domain-specific expertise beyond the built-in capabilities of LLMs, we propose **Co**gnitive **Tool**-Assisted Reasoning (**CoTool**). CoTool empowers the LLM with the autonomy to decide whether to continue internal inference or invoke an external tool at each reasoning step. The pipeline is illustrated in Algorithm 1, and detailed instructions are provided in App. B.2. Specifically, during the generation of the reasoning chain $R$, the LLM autonomously decides at each step whether to proceed with internal reasoning or invoke an external tool. At the $i$-th tool-assisted step, *i.e.*, the $i$-th step at which tool usage is deemed necessary, the LLM generates a tool query $q_{\text{tool}}^{(i)}$, enclosed between special tokens `<|begin_tool_query|>` and `<|end_tool_query|>`. Each tool query is generated based on the current state of the reasoning process and the previously collected information:

$$P(q_{\text{tool}}^{(i)}|I, q, R^{(i-1)}) = \prod_{t=1}^{T_q^{(i)}} P\left(q_{\text{tool},t}^{(i)}|q_{\text{tool},<t}^{(i)}, I, q, R^{(i-1)}, T_{\text{results}}\right), \tag{9}$$

where $I$ is the task instruction, $T_q^{(i)}$ is the length of the $i$-th tool query, $q_{\text{tool},t}^{(i)}$ is the token generated at step $t$ of the $i$-th tool query, $R^{(i-1)}$ is all prior reasoning steps before the $i$-th tool invocation, and $T_{\text{results}}$ is the result of tool query. Once the LLMs emit a tool query (*i.e.,* the special token pair `<|begin_tool_query|>` and `<|end_tool_query|>` is detected), the generation process is paused. The extracted query $q_{\text{tool}}^{(i)}$ is then executed by an external tool to obtain the $T_{\text{results}}$. The LLM then processes all the useful information to generate its subsequent reasoning and injects it back into the reasoning chain $R^{(i-1)}$, enclosed by `<|begin_tool_result|>` and `<|end_tool_result|>`. By interleaving tool usage in this manner, the model is able to resume reasoning with an enriched context that incorporates necessary information. This mechanism allows the agent to dynamically and efficiently integrate tool-assisted information into its CoT, enhancing its capability to solve complex tasks. More details in App. D.1, and the external tools it utilizes are curated from our **RSTKit toolkit** (see App. D.2).

Table 1: Accuracy (%) of baselines and the CogER on ID and OOD tasks. The 'DS-R1-DQ' is DeepSeek-R1-Distilled-Qwen2.5, and Math-72B is Qwen2.5-Math-72B-Instruct.

| Baseline | In-Domain | | | | | Out-of-Domain | | |
|---|---|---|---|---|---|---|---|---|
| | GSM8K | MATH | Com-QA | MedQA | AVG | MAWPS | College | AVG |
| Math-72B | 95.77$_{\pm 0.06}$ | 86.08$_{\pm 1.33}$ | 76.47$_{\pm 1.23}$ | 52.40$_{\pm 0.92}$ | 77.68$_{\pm 1.02}$ | 96.10$_{\pm 0.85}$ | 74.43$_{\pm 0.92}$ | 85.27$_{\pm 0.89}$ |
| DS-R1-DQ-7B | 88.12$_{\pm 1.22}$ | 89.61$_{\pm 0.46}$ | 56.07$_{\pm 1.81}$ | 26.79$_{\pm 2.44}$ | 65.15$_{\pm 1.65}$ | 91.57$_{\pm 0.41}$ | 71.96$_{\pm 0.49}$ | 81.77$_{\pm 0.45}$ |
| DS-R1-DQ-14B | 94.35$_{\pm 0.40}$ | 90.93$_{\pm 0.31}$ | 66.39$_{\pm 1.67}$ | 42.91$_{\pm 1.73}$ | 73.65$_{\pm 1.23}$ | 90.92$_{\pm 0.52}$ | 71.73$_{\pm 0.81}$ | 81.33$_{\pm 0.68}$ |
| DS-R1-DQ-32B | 95.21$_{\pm 0.35}$ | 90.73$_{\pm 0.95}$ | 59.19$_{\pm 1.90}$ | 43.29$_{\pm 1.71}$ | 72.11$_{\pm 1.75}$ | 92.37$_{\pm 0.40}$ | 72.93$_{\pm 0.51}$ | 82.65$_{\pm 0.46}$ |
| DeepSeek-R1 | **97.04**$_{\pm 0.16}$ | **96.79**$_{\pm 0.30}$ | 78.00$_{\pm 0.58}$ | 54.38$_{\pm 2.12}$ | 81.55$_{\pm 1.11}$ | 93.29$_{\pm 0.80}$ | 72.70$_{\pm 1.39}$ | 83.00$_{\pm 1.13}$ |
| L1-MAX | 92.45$_{\pm 4.53}$ | 86.33$_{\pm 1.53}$ | 48.05$_{\pm 0.09}$ | 21.97$_{\pm 0.05}$ | 62.20$_{\pm 2.39}$ | 89.90$_{\pm 3.29}$ | 63.36$_{\pm 2.29}$ | 76.63$_{\pm 2.83}$ |
| S1-32B | 94.84$_{\pm 0.40}$ | 81.07$_{\pm 14.96}$ | 75.16$_{\pm 12.11}$ | 64.14$_{\pm 1.01}$ | 78.80$_{\pm 9.64}$ | 96.78$_{\pm 0.24}$ | 73.24$_{\pm 7.21}$ | 81.32$_{\pm 5.10}$ |
| ReasonFlux-32B | 93.65$_{\pm 0.33}$ | 77.32$_{\pm 14.70}$ | 53.18$_{\pm 1.44}$ | 49.88$_{\pm 0.64}$ | 68.51$_{\pm 7.39}$ | 93.67$_{\pm 1.08}$ | 78.83$_{\pm 6.13}$ | 86.25$_{\pm 4.40}$ |
| RouteLLM | 95.80$_{\pm 0.05}$ | 87.29$_{\pm 0.12}$ | 83.99$_{\pm 0.19}$ | 79.22$_{\pm 0.32}$ | 86.58$_{\pm 0.20}$ | **97.90**$_{\pm 0.00}$ | 87.29$_{\pm 0.08}$ | 92.60$_{\pm 0.06}$ |
| **CogER (Ours)** | 96.18$_{\pm 0.05}$ | 95.20$_{\pm 0.20}$ | **84.52**$_{\pm 0.30}$ | **81.23**$_{\pm 0.00}$ | **89.28**$_{\pm 0.18}$ | 97.87$_{\pm 0.01}$ | **89.24**$_{\pm 0.14}$ | **93.56**$_{\pm 0.10}$ |

### 4.5 TRAINING WITH GROUP RELATIVE POLICY OPTIMIZATION

We adopt the Group Relative Policy Optimization (GRPO) (Shao et al., 2024; Guo et al., 2025) to optimize the parameters $\theta$ of the CogER-Agent due to its superior stability and sample-efficiency.

**Group Relative Advantage Estimation.** For each query $x$, a group of $G$ candidate outputs $\{o_1, o_2, \ldots, o_G\}$ is sampled from the old policy model $\pi_{\theta_{old}}$. Each output is then scored according to the reward function defined in Eqn. (3), yielding a set of rewards $r = \{r_1, r_2, \ldots, r_G\}$. Subsequently, these rewards are normalized by subtracting the group mean and dividing by the group standard deviation. The normalized reward $\tilde{r}_i = \frac{r_i - \text{mean}(r)}{\text{std}(r)}$ is then used as outcome supervision. Specifically, the normalized reward $\tilde{r}_i$ is assigned as the advantage $\hat{A}_i$ to all tokens within the corresponding output $o_i$, $i.e.,$ $\hat{A}_i = \tilde{r}_i$. The policy is then updated by maximizing the objective.

**Learning Objectives.** The goal of the learning is to maximize the expected long-term return $\mathcal{J}(\theta)$:

$$\mathcal{J}_{GRPO}(\theta) = \mathbb{E}[x \sim P(Q), \{o_i\}_{i=1}^G \sim \pi_{\theta_{old}}(O|x)]$$

$$\frac{1}{G} \sum_{i=1}^{G} \left\{ \min \left[ \frac{\pi_\theta(o_i|x)}{\pi_{\theta_{old}}(o_i|x)} \hat{A}_i, \text{clip} \left( \frac{\pi_\theta(o_i|x)}{\pi_{\theta_{old}}(o_i|x)}, 1 - \varepsilon, 1 + \varepsilon \right) \hat{A}_i \right] - \beta \mathrm{D}_{KL} [\pi_\theta || \pi_{ref}] \right\},$$

$$(10)$$

where $\varepsilon$ and $\beta$ are hyper-parameters, $\pi_\theta$ and $\pi_{\theta_{old}}$ are the current and old policy models.

## 5 EXPERIMENTS

**Datasets and Metrics.** To train the CogER-Agent, we construct the *Reasoning-Training dataset* by randomly sampling 2,000 examples from each of four heterogeneous benchmarks: GSM8K (Cobbe et al., 2021), MATH (Hendrycks et al., 2021), CommonsenseQA (Talmor et al., 2019), and MedQA (Jin et al., 2021). This unified training set exposes the agent to a wide spectrum of reasoning challenges, from arithmetic word problems to domain-specific medical questions. For evaluation, we consider both In-Domain (ID) and Out-of-Domain (OOD) settings. ID performance is evaluated on the official test splits of GSM8K, MATH-500, CommonsenseQA, and MedQA, whereas OOD generalization is measured on MAWPS (Koncel-Kedziorski et al., 2016) and CollegeMath (Tang et al., 2024), which are not included in the mixed training set used to fine-tune our CogER-Agent. More details in App. C. We report Exact Match ($EM$) as the metric across all datasets, and record the average parameters (Param.) and Latency used during testing to reflect computational cost.

**Baselines.** We employ LLMs with varying sizes and architectures, including Qwen2.5-Math-72B-Instruct (Yang et al., 2024b), DeepSeek-R1 (Guo et al., 2025), DeepSeek-R1-Distill-Qwen-7B, DeepSeek-R1-Distill-Qwen-14B, and DeepSeek-R1-Distill-Qwen-32B. We compare against Test-Time Scaling (TTS) methods, including S1 (Muennighoff et al., 2025), L1 (Aggarwal & Welleck, 2025), and ReasonFlux (Yang et al., 2025a). In addition, we include the LLM routing method RouteLLM (Ong et al., 2025b) as a representative routing-based baseline.

**Implementation Details.** In our CogER framework, Qwen2.5-7B-Instruct (Yang et al., 2024a) serves as the CogER-Agent, which assigns queries to appropriate reasoning modules based on their

Table 2: Accuracy (%) of each reasoning mode and the proposed CogER on ID and OOD tasks.

| Version | ID | OOD |
|---|---|---|
| Oracle | 94.85 | 96.61 |
| $L_1$ (Qwen2.5-7B-Instruct) | 76.28 | 86.23 |
| $L_2$ (Qwen2.5-32B-Instruct) | 83.62 | 89.49 |
| $L_3$ (QWQ-32B) | 86.75 | 93.13 |
| $L_4$ (Our CoTool) | 88.42 | 92.89 |
| **CogER (Ours)** | **89.28** | **93.56** |

Table 3: Results for the component of the reward function. "w/o" denotes the removal of the specified reward term.

| Version | ID | OOD |
|---|---|---|
| Training-free | 86.35 | 92.78 |
| w/o $\mathcal{R}_{\text{format}}$ | 87.37 | 93.42 |
| w/o $\mathcal{R}_{\text{hierarchy}}$ | 87.89 | 92.21 |
| **CogER (Ours)** | **89.28** | **93.56** |

Table 4: Proportion of queries routed to each complexity level by the CogER-Agent, with and without the fallback-reward component $\mathcal{R}_{\text{hierarchy}}$.

| Level | $L_1$ | $L_2$ | $L_3$ | $L_4$ |
|---|---|---|---|---|
| w/o $\mathcal{R}_{\text{hierarchy}}$ | 2.32% | 8.30% | 0.92% | 88.46% |
| **CogER (Ours)** | 2.00% | 28.17% | 21.90% | 47.93% |

Table 5: Impact of CoTool on $EM$ and Tool Invocation Rate ($TIR\%$).

| Version | MATH-500 | | CollegeMath | |
|---|---|---|---|---|
| | $EM$ | $TIR$ | $EM$ | $TIR$ |
| w/o CoTool | 87.20 | - | 87.93 | - |
| CoTool | **97.00** | 3.03 | **89.04** | 5.17 |

estimated complexity, and also handles all $L_1$-level queries directly. Queries classified as $L_2$-level are escalated to Qwen2.5-32B-Instruct for moderate multi-step reasoning, while $L_3$-level queries are processed by QwQ-32B (Team, 2025) to support deeper CoT generation. For the most demanding $L_4$-level queries, we invoke our CoTool, whereby QwQ-32B (Team, 2025) autonomously issues external API calls to enrich its reasoning process. We uniformly capped the generation length at **max_token = 8192** for all LLMs. Furthermore, all components are optimized using the AdamW optimizer with a batch size of $24 \times 3$ and a learning rate of $5 \times 10^{-5}$. The group size $G$ in Eqn. (10) is set to 12. The CogER-Agent is fine-tuned via LoRA with a rank of $r = 16$, while all other hyperparameters follow the default settings from the Open-R1 configuration (Face, 2025). More details can be found in App. D.3.

## 5.1 COMPARISON EXPERIMENTS

To evaluate the effectiveness of our CogER, we compare it against several baselines, including the original LLM, L1-MAX, S1-32B, ReasonFlux-32B, and RouteLLM. Results are in Table 1.

**Superior performance on ID tasks.** From Table 1, our CogER achieves the best performance on ID tasks. Specifically, compared to DeepSeek-R1, CogER achieves a **relative** performance **improvement** of 9.48% (81.55 → 89.28) in terms of average $EM$ metric. Notably, CogER outperforms generic LLMs on knowledge-intensive benchmarks. Our CogER consistently outperforms the SOTA TTS methods. For example, compared with S1-32B, our CogER has a relative improvement of 13.30% in terms of average $EM$ metric. Moreover, compared with RouteLLM, CogER further improves the average $EM$ from 86.58 to 89.28. This is primarily attributed to its ability to route each query to the most suitable reasoning strategy, thereby leveraging the strengths of different models.

**Superior performance on OOD tasks.** To assess the generalization ability of our CogER beyond the training distribution, we conduct experiments on MAWPS and CollegeMath. From Table 1, CogER achieves an average $EM$ accuracy of 93.56%, consistently outperforming both the original LLMs and SOTA TTS methods. Specifically, on the MAWPS dataset, our method achieved a relative improvement of 1.84% and 1.13% over Qwen2.5-Math-72B-Instruct and S1-32B, respectively. On the more challenging CollegeMath dataset, CogER achieves 89.24%, with substantial relative improvements of 13.21% over ReasonFlux-32B. Moreover, compared with the routing baseline RouteLLM, CogER achieves a higher average $EM$ on OOD tasks (93.56 vs. 92.60). These results demonstrate that CogER effectively adapts its reasoning strategies to unseen data by leveraging its complexity-aware routing mechanism.

## 5.2 ABLATION STUDIES

**Effectiveness of CogER.** We compare CogER against each standalone reasoning strategy. From Table 2, CogER outperforms all single-strategy baselines, achieving 89.28% $EM$ on ID tasks and 93.56% $EM$ on OOD tasks. Moreover, Table 4 presents the distribution of reasoning actions se-

lected by our CogER-Agent. Note that as CogER-Agent acts as both router and $L_1$ solver, its problem-solving ability slightly degrades after training, leading to a lower $L_1$ share that is expected and by design. The relatively balanced selection across strategies indicates that the agent learns to exploit the complementary strengths of different reasoning modes, rather than relying heavily on any single one. These findings highlight that dynamically routing queries based on task complexity leads to more robust and accurate reasoning than any fixed, one-size-fits-all approach.

**Effectiveness of RL Training.** To evaluate the RL training strategy, we compare CogER with a training-free prompt engineering baseline. From Table 3, CogER outperforms the training-free baseline, yielding a relative improvement of 3.39% on ID tasks and 0.84% on OOD tasks. These results demonstrate that learning to adaptively select strategies via reinforcement learning is not only more effective, but also more robust and generalizable than static, training-free alternatives.

**Impact of the reward function $\mathcal{R}$.** We investigate the effects of Format Reward $\mathcal{R}_{\text{format}}$ and Hierarchical-Aware Reward $\mathcal{R}_{\text{hierarchy}}$ on the performance of CogER. From Table 3, removing the Format Reward $\mathcal{R}_{\text{format}}$ results in a noticeable performance drop on both ID ($89.28 \to 87.37$) and OOD ($93.56 \to 93.42$) tasks, indicating that this reward is essential for guiding the CogER-Agent to select appropriate reasoning strategies reliably. Removing the Hierarchical-Aware Reward not only leads to overall performance degradation, but also causes the agent to excessively favor the $L_4$ (Delegate) strategy (88.46%), as reported in Table 4, resulting in unnecessary computational cost.

**Effectiveness of CoTool.** We compare model performance with and without CoTool on both ID and OOD tasks. From Table 5, integrating CoTool leads to a relative improvement of 11.24% in $EM$ on ID tasks ($87.20 \to 97.00$) with only 3.03% tool invocation, and $EM$ on OOD is improved by 1.26% ($87.93 \to 89.04$) with a tool invocation rate of 5.17%. These results suggest that CoTool effectively enhances the model's ability to handle complex queries by selectively leveraging external tools.

## 5.3 MORE DISCUSSIONS

**Cross-task generalization.** To examine whether CogER overfits specific math and commonsense benchmarks, we further evaluate it on three unseen tasks with very different formats: MBPP (Austin et al., 2021) for code generation, QuALITY (Pang et al., 2022) for long-context multiple-choice QA, and Natural Questions (Kwiatkowski et al., 2019) for retrieval-augmented factual QA. As shown in Table 6, CogER achieves the best or tied-best performance on all three benchmarks, attaining 91.76 Pass@3 on MBPP, matching the top QuALITY accuracy of 82.97, and obtaining the highest F1-score of 67.25 on Natural Questions, while using comparable or lower computational cost than strong baselines such

Table 6: Performance of CogER and baselines on additional benchmarks, including code generation (MBPP, Pass@3), long-context multiple-choice QA (QuALITY, accuracy), and retrieval-augmented factual QA (Natural Questions, F1-score).

| Baseline | MBPP | QuALITY | Natural Questions |
|---|---|---|---|
| Math-72B | 75.16 | 55.77 | 45.52 |
| DS-R1-DQ-7B | 77.30 | 35.21 | 15.41 |
| DS-R1-DQ-14B | 79.88 | 73.33 | 61.14 |
| DS-R1-DQ-32B | 91.22 | 81.22 | 64.45 |
| L1-MAX | 38.30 | 25.45 | 2.96 |
| S1-32B | 86.00 | 81.41 | 66.99 |
| ReasonFlux-32B | 91.70 | 81.88 | 63.37 |
| RouteLLM | 90.68 | **82.97** | 67.06 |
| **CogER(Ours)** | **91.76** | **82.97** | **67.25** |

as DS-R1-DQ-32B and RouteLLM. These results indicate that the proposed cognitive hierarchy and elastic level selection transfer beyond the GSM8K/MATH/Com-QA training domains and remain effective on diverse, out-of-distribution reasoning tasks.

**Compute efficiency.** We analyze the computational cost of CogER, TTS methods, and the best-performing LLM, DeepSeek-R1, to substantiate the computational efficiency of the proposed method. From Table 7, CogER achieves the lowest end-to-end latency (118.53s) and the fewest generated words (489.71) with an effective participating scale of 29.6B parameters. Specifically, CogER achieves SOTA accuracy while reducing latency by 76.58% (over 4 times faster) compared to the top-performing baseline (DeepSeek-R1). These results support our claim that a complexity-aware CogER-Agent yields computational savings while preserving the accuracy gains.

**Impact of different routing strategies.** We study the impact of different query selection strategies on overall performance. From Table 8, a four-class classifier underperforms the CogER, indicating that purely supervised routing is insufficient to capture the uncertainty of query difficulty. Modeling routing as an MDP and training with RL enables exploration and credit assignment over sequences,

Table 7: Computational cost averaged over all datasets. Parameters (Param.), latency (s), and generated words per query are reported. * is the latency of the CogER-Agent for generating the level tag.

| Baseline | Param. ↓ | Latency ↓ | Words ↓ |
|---|---|---|---|
| QWQ-32B | 32B | 147.21 | 1160.67 |
| DeepSeek-R1 | 671B | 506.19 | 654.63 |
| L1-MAX | **1.5B** | 190.14 | 1149.68 |
| S1-32B | 32B | 273.47 | 946.70 |
| ReasonFlux-32B | 32B | 286.97 | 1050.63 |
| **CogER (Ours)** | 29.6B | **118.53 (0.01*)** | **489.71** |

Table 8: Comparison of different query selection strategies on ID and OOD tasks. Random denotes uniform sampling over reasoning strategies, and Classifier corresponds to a flat four-class classifier (router) trained to predict query levels.

| Version | ID | OOD |
|---|---|---|
| Random | 84.21 | 90.28 |
| Classifier | 84.09 | 90.32 |
| **CogER (Ours)** | **89.28** | **93.56** |

allowing the agent to discover non-myopic policies that allocate computation adaptively. Consequently, CogER attains higher $EM$ on both ID and OOD settings than Random and Classifier.

## 6 CONCLUSION

In this paper, we have proposed **Cog**nitive-Inspired **E**lastic **R**easoning (**CogER**), a dynamic reasoning framework designed to address the challenge of handling queries with varying complexity in a cost-effective and accurate manner. Inspired by **Bloom's Taxonomy**, CogER first assesses the complexity of each input query and assigns it to one of four cognitive levels, each corresponding to a distinct reasoning strategy. To dynamically select the most appropriate strategy, we formulate the selection process as an MDP and train a CogER-Agent via RL. The agent is guided by a reward function that balances solution quality with computational efficiency, ensuring that complex queries receive sufficient reasoning depth while simpler ones are handled with minimal overhead. Moreover, for $L_4$ queries requiring external knowledge or specialized capabilities, we introduce a Cognitive Tool-Assisted Reasoning that enables the agent to autonomously invoke external tools within its CoT when necessary, enhancing its ability to address not only knowledge-intensive queries, but also those involving structured data retrieval, numerical reasoning, or factual verification. Extensive experiments demonstrate that CogER significantly outperforms SOTA TTS methods, achieving a 13% relative improvement in average exact match on ID tasks and an 8% gain on OOD tasks.

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

# Supplementary Materials for
# "Beyond Fast and Slow: Cognitive-Inspired Elastic Reasoning for Large Language Models"

CONTENTS

# A MORE RELATED WORK

## A.1 LARGE LANGUAGE MODELS

Recent advancements in Natural Language Processing (NLP) have culminated in the development of highly capable Large language models (LLMs). These models are predominantly characterized by their utilization of the Transformer architecture (Vaswani et al., 2017) and are pre-trained on vast quantities of textual data. Progressive scaling of model parameters and training datasets has allowed these LLMs to exhibit emergent capabilities (Wei et al., 2022a), demonstrating remarkable proficiency in the handling of a variety of complex tasks. Such tasks include, but are not limited to, high-fidelity question answering (Shao et al., 2023), code generation (Chen et al., 2021), and intermediate-step reasoning (Wei et al., 2022b). Consequently, LLMs have exerted a profound influence on the Artificial Intelligence (AI) community, catalyzing a reevaluation of the prospects for Artificial General Intelligence (AGI) (Zhao et al., 2023). Predicated on their foundational Transformer architecture, extant Large language models can be broadly classified into three primary categories:

**Encoder-only models.** Encoder-only models, alternatively designated as auto-encoding architectures, are exclusively constructed from encoder modules originating from the Transformer architecture. In such configurations, input data is processed sequentially across multiple layers, facilitating the progressive extraction and encoding of information. A paradigmatic example of this category is BERT (Bidirectional Encoder Representations from Transformers) (Devlin et al., 2018), developed by Google. BERT functions as a language representation model employing bidirectional Transformer encoders. It was pre-trained on a corpus consisting of the BooksCorpus (Zhu et al., 2015) (comprising approximately 800 million words) and English Wikipedia (approximately 2.5 billion words). This pre-training enabled BERT to attain a score of 80.5% on the General Language Understanding Evaluation (GLUE) benchmark and an accuracy of 86.7% on the Multi-Genre Natural Language Inference (MultiNLI) task. Many subsequent encoder-only models are predominantly variants of BERT, including RoBERTa by Meta (Liu et al., 2019) and DeBERTa by Microsoft (He et al., 2020).

**Decoder-only models.** This category of models is exclusively constructed using decoder modules from the Transformer architecture. Decoder-only models typically implement an auto-regressive mechanism, whereby output sequences are generated on a token-by-token basis. The generation of each token by the decoder is contingent upon the tokens previously generated. Seminal examples in this category include the Generative Pre-trained Transformer (GPT) series (Achiam et al., 2023), developed by OpenAI. Illustratively, GPT-3 comprises numerous Transformer decoder layers, featuring up to 175 billion parameters, which established it as one of the most substantial language models at its introduction. This model was pre-trained on a corpus of approximately 300 billion tokens derived from sources such as Common Crawl (Raffel et al., 2020), WebText2, Books1, Books2, and Wikipedia. GPT-3 has demonstrated potent zero-shot and few-shot learning capabilities across a diverse array of language tasks. Beyond the GPT series, a multitude of other decoder-only models have emerged, including OPT (Zhang et al., 2022), LLaMA (Touvron et al., 2023a), and Llama 2 (Touvron et al., 2023b) from Meta; PaLM (Chowdhery et al., 2023) and PaLM 2 (Anil et al., 2023) from Google; and BLOOM (Workshop et al., 2022) from the BigScience initiative. Furthermore, models such as Qwen 2.5 by Alibaba (Yang et al., 2024a), alongside DeepSeek LLM (Bi et al., 2024) and DeepSeek Coder (Guo et al., 2024) by DeepSeek AI, continue to advance the state-of-the-art in language comprehension, multilingual processing, and domain-specific generation.

**Encoder-decoder models.** This architectural class integrates both encoder and decoder modules from the Transformer framework. Such models aim to amalgamate the respective strengths of the aforementioned architectures, thereby effectively addressing tasks that necessitate comprehensive input understanding coupled with the generation of extended output sequences. Prominent extant encoder-decoder models encompass GLM from Tsinghua University (Du et al., 2022); T5 (Raffel et al., 2020), FLAN-T5 (Chung et al., 2024), and UL2 (Tay et al., 2022) from Google; and BART (Lewis et al., 2019) from Meta. For instance, the GLM model employs an autoregressive blank infilling objective. This methodology is designed to tackle three fundamental challenges in NLP: natural language understanding (NLU), unconditional text generation, and conditional text generation. With a reported maximum of 130 billion parameters, GLM was pre-trained on a corpus including BookCorpus (Tay et al., 2022) and Wikipedia. GLM has demonstrated superior performance over BERT on the SuperGLUE benchmark, exhibiting an improvement of 4.6%–5.0%. Moreover, it signifi-

cantly surpasses FLAN-T5 in both NLU and generation tasks, even when utilizing fewer parameters and less training data.

## A.2 LARGE REASONING MODELS

Large language models (LLMs) have demonstrated remarkable capabilities in natural language understanding and complex reasoning (Grattafiori et al., 2024), becoming a pivotal advancement in AI. To further enhance performance in "System 2" reasoning domains (Li et al., 2025) such as mathematics (Cobbe et al., 2021; Hendrycks et al., 2021) and programming (Chen et al., 2021), which require deep thought, researchers have developed specialized Large Reasoning Models (LRMs) (Xu et al., 2025a).

**Fundamentals of Large Reasoning Models.** The core of LRMs lies in internalizing and enhancing Chain-of-Thought (CoT) (Wei et al., 2022b) reasoning through strategies like Supervised Fine-Tuning (SFT) and Reinforcement Learning (RL). CoT significantly improves LLM performance on complex tasks by prompting the model to generate a series of intermediate reasoning steps, with variants like Self-Consistency CoT (Wang et al., 2023), Tree-of-Thoughts (Yao et al., 2023), and Graph-of-Thoughts (Besta et al., 2024) emerging. LRMs aim to integrate this step-by-step reasoning capability more deeply within the model, rather than relying solely on explicit test-time prompts or external augmentations, by generating detailed, structured reasoning sequences to achieve higher accuracy. Prominent examples of LRMs include OpenAI's o1 model and DeepSeekAI's DeepSeek-R1 model.

**Training Mechanisms of LRMs.** The training mechanisms for LRMs typically combine SFT to learn from high-quality reasoning paths and RL to further optimize reasoning strategies, enabling exploration of better problem-solving steps (Luo et al., 2025; Aggarwal & Welleck, 2025). For instance, DeepSeek-R1 enhanced its general reasoning capabilities through multiple rounds of SFT and RL, emphasizing structured thinking templates and rule-based reward mechanisms. OpenAI's o1 is speculated by the community to employ tree-search methods like Monte Carlo Tree Search (MCTS) (Coulom, 2006) combined with a Process Reward Model (PRM) (Uesato et al., 2022) to explore and evaluate different reasoning paths. These advanced training methods enable LRMs to generate complex thought processes internally, progressively deriving final answers, demonstrating significant potential in solving challenging mathematical problems and programming tasks, as assessed by benchmarks like Sys2Bench (Parashar et al., 2025).

## A.3 REINFORCEMENT LEARNING

Reinforcement Learning (RL) (Kaelbling et al., 1996) constitutes a paradigm within machine learning wherein an agent learns to optimize its decision-making process through interaction with an environment. This interaction involves performing actions and receiving consequent feedback, typically in the form of rewards or penalties. The principal learning objective in RL is the maximization of a cumulative reward signal. In contrast to supervised learning, which relies on datasets comprising pre-defined input-output pairs for model training, RL entails an agent acquiring knowledge from the repercussions of its actions, mediated by this reward-penalty mechanism. This iterative, trial-and-error learning process, coupled with its emphasis on sequential decision-making under uncertainty, distinguishes RL from supervised learning methodologies that depend on labeled datasets. Existing reinforcement learning algorithms can be broadly categorized based on whether an explicit model of the environment is learned or utilized, leading to two principal classes: Model-free RL and Model-based RL.

**Model-free RL.** Model-free RL algorithms enable the agent to learn optimal policies directly from trajectory samples accrued through interaction with the environment, without explicitly constructing an environmental model. Within model-free RL, algorithms are further distinguished by the components they learn, leading to three primary sub-categories: actor-only, critic-only, and actor-critic algorithms. Actor-only algorithms directly learn a policy network, denoted as $\pi_\theta(a|s)$, which maps states to actions. This network takes the current state $s_t$ as input and outputs the action $a_t$. Prominent examples of such algorithms include Reinforce (Williams, 1992) and various policy gradient methods (Sutton et al., 1999). Critic-only algorithms, in contrast, focus solely on learning a value function (e.g., state-value or action-value function). Given a state $s_t$, the learned value model is used to evaluate all possible actions $a' \in A$, and the action $a_t$ yielding the maximum estimated

value is selected. This category encompasses methods such as Q-learning (Watkins, 1989). Actor-critic algorithms combine these two approaches by concurrently maintaining and learning both a policy network (the actor) for action selection and a value function model (the critic) for evaluating actions or states. This category includes algorithms such as Deep Deterministic Policy Gradient (DDPG) (Lillicrap et al., 2015), Trust Region Policy Optimization (TRPO) (Schulman et al., 2015), Proximal Policy Optimization (PPO) (Schulman et al., 2017), and Asynchronous Advantage Actor-Critic (A3C) (Mnih et al., 2016). Notably, PPO has gained considerable traction for training large language models. Recent advancements in this area include GRPO (Zhang et al., 2024), which employs group-based advantage estimates within a KL-regularized loss function to reduce computational overhead and enhance update stability, and DAPO (Chen et al., 2024a), which utilizes distinct clipping mechanisms and adaptive sampling techniques to improve efficiency and reproducibility during the fine-tuning of large-scale models.

**Model-based RL.** Model-based RL algorithms endeavor to learn an explicit model of the environment, thereby addressing challenges related to sample efficiency. This is because the agent can leverage the learned model for planning and decision-making, reducing the necessity for extensive direct environmental interaction. The learned representation of the environment is commonly termed a 'world model'. This world model typically predicts the subsequent state $s_{t+1}$ and the immediate reward $r_t$ based on the current state $s_t$ and the action $a_t$ taken. Exemplary model-based RL algorithms include Dyna-Q (Peng et al., 2018), Model-Based Policy Optimization (MBPO) (Janner et al., 2019), and Adaptation Augmented Model-based Policy Optimization (AMPO) (Shen et al., 2023).

### A.4 TOOL INTEGRATED REASONING

Research in Tool-Integrated Reasoning (TIR) aims to enhance the capabilities of large language models (LLMs) by enabling them to effectively utilize external tools for complex problem-solving. The related literature can be broadly categorized as follows:

**Foundations and Evaluation of Tool-Integrated Reasoning.** Early research predominantly focused on equipping LLMs with external tools to overcome their inherent limitations. This involved introducing concepts such as program executors (Chen et al., 2023) and search engines (Vu et al., 2024) to enhance their problem-solving capabilities (Qin et al., 2024a). The core tenet of TIR is to enable LLMs to interact with these external tools, thereby addressing issues such as outdated knowledge, computational inaccuracies, and shallow reasoning (Qian et al., 2025a). As research progressed, a series of specialized benchmarks were proposed to systematically evaluate model performance in tool selection, argument generation, and generalization (Qin et al., 2024b). Concurrently, the construction of high-quality tool-use datasets became a significant driver for advancements in the field (Liu et al., 2024; Qian et al., 2025b), and these datasets and benchmarks have further facilitated the exploration of TIR techniques across diverse modalities and specialized domains (Shen et al., 2025).

**Supervised Fine-Tuning for Tool-Integrated Reasoning.** In the initial stages of training LLMs for TIR tasks, Supervised Fine-Tuning (SFT) was the predominant approach. These methods typically relied on offline-generated tool-use trajectories, upon which models were subsequently fine-tuned (Chen et al., 2024c; Acikgoz et al., 2025). For instance, in code-integrated reasoning scenarios, researchers endowed models with initial capabilities for code invocation and result interpretation by performing SFT on self-curated Chain-of-Thought (CoT) data that included code execution steps (Chen et al., 2025b). However, SFT methods exhibit notable deficiencies in terms of generalization, exploration, and adaptability (Chu et al., 2025; Guo et al., 2025). Models often merely imitate specific patterns within the training data, struggling to adapt to unseen or more complex tool-use scenarios and failing to autonomously learn when and how to invoke external tools most effectively (Feng et al., 2025).

**Reinforcement Learning for Tool-Integrated Reasoning.** To overcome the limitations inherent in SFT, Reinforcement Learning (RL) has emerged as a promising paradigm for training more adaptive and generalizable tool-using LLMs. RL frameworks enable models to learn optimal tool invocation strategies through direct interaction with environments and feedback signals, moving beyond the mere imitation of static trajectories. Various RL algorithms, including Proximal Policy Optimization (PPO) and Direct Preference Optimization (DPO) , are being adapted to the specific challenges

of TIR. Central to applying RL in TIR is the development of effective feedback mechanisms. One stream of research focuses on meticulous reward engineering, designing rewards that offer step-grained guidance on tool invocation correctness and contribution (Yu et al., 2024), or penalize specific error types (Ye et al., 2024). Another prominent trend involves learning from preferences or comparative feedback, often leveraging techniques like Direct Preference Optimization (DPO) or ranking losses. This allows models to learn from a broader spectrum of execution traces, including imperfect or erroneous paths, by comparing preferred outcomes against less desirable ones (Chen et al., 2024b; Zeng et al., 2025; Jung et al., 2025). Such approaches also extend to optimizing multi-turn dialogue control and learning from varied forms of execution feedback (Wu et al., 2024). Collectively, these RL strategies aim to enhance not only the precision of tool selection and usage but also the model's nuanced decision-making in complex, interactive scenarios.

## B    INSTRUCTION TEMPLATES

### B.1    SYSTEM PROMPT FOR COGER

This system prompt is designed to guide the 7B model (i.e., the CogER-Agent), fine-tuned using GRPO LoRA. Its primary function is to analyze user queries and accurately classify them into one of four predefined complexity levels ($L_1$ to $L_4$), laying the groundwork for the subsequent differentiated processing pipeline.

---

**System Prompt for CogER:**

You are a helpful AI Assistant that classifies and solves user queries based on their complexity level. Your task is to analyze a given question, classify it into one of the following levels (L1-L4). For questions at level L1, you also need to directly provide the answer.
Classify the input question into L1-L4 based on the criteria.
L1 level: These are straightforward questions that require no external tools or deep reasoning. You can answer these questions directly.
L2 level: These questions require logical reasoning and the ability to make inferences but do not need external tools. Your answer will involve some reasoning steps before arriving at the conclusion.
L3 level: These questions involve a more extended chain of thought or multiple sub-steps to reach an answer. Your reasoning process will be more involved but still remains independent of external tools.
L4 level: These questions need external resources or tools to complete. You will need to incorporate tool calls to provide a comprehensive response.
Instructions: 1) For L1 questions: Directly output the answer followed by the level, in the format `<question_level>`L1`</question_level>`.
2) For L2-L4 questions:    Output the corresponding level only, enclosed within `<question_level>` tags, with LEVEL replaced by L2/L3/L4.
3) Never explain your classification logic. If uncertain, choose the higher level (e.g., borderline L1/L2 to L2).

---

### B.2    INSTRUCTION FOR COTOOL

**Main Instruction for CoTool.** This instruction empowers the LLM when processing L4-level complex queries, guiding it to autonomously identify needs, formulate tool queries, and interact with external tools when external knowledge or computational capabilities are required, while managing the entire reasoning process until a final answer is generated.

Instruction for CoTool:

You are a professional reasoning expert tasked with accurately answering the user's question. Your reasoning process should be step-by-step and transparent.

When your internal knowledge is insufficient, or when you require specific, up-to-date information (like real-time data or complex calculations) to proceed accurately, you must identify the precise information needed and invoke an external tool to retrieve or compute it.

Available Tool Categories:

To assist you, a suite of tools is available. While the system will automatically select the most appropriate specific tool (e.g., Weather API, Search Engine, Calculator) based on your query, you should understand the task categories these tools can handle:

- Information Retrieval:
- Fetching real-time data (e.g., current weather, stock prices).
- Searching the web for specific facts or general knowledge.
- Extracting information from documents (e.g., PDFs).

- Calculation & Symbolic Math:
- Performing arithmetic operations, etc.
- Solving algebraic equations, etc.
- Calculating derivatives, integrals, limits, etc.
- Performing statistical calculations.

How to Use Tools:

1. Identify Need: In your reasoning steps, clearly state what specific information is missing or what needs verification/calculation.

2. Formulate Query: Based on the identified need, formulate a concise query for the information retrieval or calculation tool.

3. Invoke Tool: Use the special marker `<|begin_tool_query|>`followed by your query, and end with `<|end_tool_query|>`.

- Format: `<|begin_tool_query|>` your concise query `<|end_tool_query|>`.

4. Receive Information/Result: The system will execute your request using the most appropriate available tool and provide the result within the `<|begin_tool_result|>` and `<|end_tool_result|>` markers.

- Format: `<|begin_tool_result|>` relevant information or result from the tool `<|end_tool_result|>`

- Note: The content inside these markers is the direct output from the tool (e.g., answer, data).

Tool Usage Limit:

You can invoke tools multiple times if necessary. However, the maximum number of tool calls allowed is {MAX_TOOL_CALLS}.

Continue Reasoning:

After receiving the tool result, integrate it into your reasoning chain and proceed towards the final answer.

Remember:

- Clearly state the reason for needing the tool before invoking it.

- Use the exact `<|begin_tool_query|>`...`<|end_tool_query|>` format.

- Integrate the tool's result (`<|begin_tool_result|>`...`<|end_tool_result|>`) into your ongoing reasoning.

- Focus on providing a final, accurate answer based on the complete reasoning chain.

Please answer the following question. You should provide your detailed final answer in the format

boxed{YOUR_DETAIL_ANSWER}.

Question:

**Task Instruction.** This instruction is utilized after an external tool has executed and returned its result. It guides the LLM in analyzing the validity of the tool's output, integrating key information into the current reasoning chain, and planning the next course of action based on this new information.

---

**Task Instruction:**

You previously decided to use a tool to answer the sub-query or perform the task: "**{tool_query}**".
The system executed this using the most appropriate tool and returned the following output. Your task is to analyze this output and determine the next step in your reasoning process to answer the original question.
Guidelines:
1. Analyze the Tool Output:
- Carefully review the output provided by the tool.
- Evaluate its relevance and usefulness specifically in relation to the task "**{tool_query}**".
- Note whether the tool execution was successful (status: success) or resulted in an error (status: error).
2. Determine Next Step:
- If the output is helpful and the tool succeeded: Integrate the key information into your reasoning. State the next logical step based on this new information.
- If the tool reported an error (status: error):Acknowledge the error in your reasoning. Decide if you need to re-phrase the tool query, try a different tool, or proceed without the tool's result.
- If the output is unhelpful or irrelevant (even if status is success): Acknowledge this. Decide whether to try a different tool query or proceed without this information.
3. Output Format:
- State your analysis of the tool output and clearly define the next step in your reasoning process.
- Do not simply repeat the tool output; explain how it affects your plan.
- Continue your step-by-step reasoning.

Inputs:
- Previous Reasoning Steps:
{prev_reasoning}
- Current Tool Query/Task Executed:
{tool_query}
- Formatted Tool Output:
{tool_output}

---

**Tool Selection Instruction.** This instruction specifically directs the model (or a dedicated tool selection module) to choose the most suitable tool from the RSTKit's repertoire based on the LLM-generated tool query, and to generate the necessary parameters for invoking that tool in a strict JSON format.

---

**Tool Selection Instruction:**

You are an expert tool selection assistant. Based on the user query and the available tools listed below, choose the single best tool to fulfill the request.

Available Tools:
{TOOL_LIST}
User Query: {TOOL_QUERY}

Instructions:
1. Analyze the user query carefully.
2. Evaluate each tool's description to see if it matches the query's intent.
3. Prioritize using specific tools (like calculators, weather tools, search engines, etc.) if they directly address the query.
4. IMPORTANT: Only choose the 'execute_generated_code' tool if none of the other available tools can not address the user's query. This tool is for complex calculations, custom logic, or tasks not covered by standard tools.
5. Provide your answer ONLY in JSON format with the fields 'tool_name' and 'parameters'.
6. Ensure the 'parameters' field contains all required parameters for the chosen tool, based on the user query.
7. Do not include any extra text, explanations, or markdown formatting. Your entire response must be a single, valid JSON object.

Response JSON format:
{{
"tool_name": "`<name_of_selected_tool>`",
"parameters": `<parameters_object>`
}}

---

## C BENCHMARKS

To comprehensively evaluate the performance of our proposed method across diverse reasoning tasks, we employed several academically recognized benchmarks, as detailed in Section 5. These benchmarks span a spectrum from fundamental arithmetic and university-level mathematics to commonsense reasoning and specialized domain knowledge. A detailed description of each benchmark, including its primary sources and licensing information, is provided below:

- **GSM8K** (Cobbe et al., 2021) is a widely adopted benchmark for evaluating the arithmetic reasoning capabilities of language models on grade-school math word problems. The dataset comprises 7,473 training examples and 1,319 test examples. Problems are designed to necessitate 2 to 8 steps of chain-of-thought reasoning for their solution. These problems are human-curated to ensure linguistic diversity and assess the model's understanding of fundamental mathematical concepts and its ability to perform multi-step arithmetic operations. The dataset is available at this link and under the MIT License.

- **MATH** (Hendrycks et al., 2021), is a challenging competition-level mathematics dataset designed to measure reasoning capabilities in advanced mathematics. It encompasses problems from pre-algebra, algebra, geometry, number theory, and calculus, among others, totaling 7,500 training examples and 5,000 test examples. These problems typically require complex symbolic manipulation, abstract thinking, and multi-step deductive reasoning. The dataset is available at this link and under the MIT License.
  For our experiments, we utilize **MATH-500**, an evaluation subset consisting of 500 problems sampled from the original MATH test set. The sampling ensures that the evaluation data maintains a distribution similar to the MATH training data, adheres to an independent and identically distributed (I.I.D.) characteristic among test samples, and has no overlap with the training set. The selection of MATH-500 balances problem difficulty and diversity with manageable computational resources and evaluation time, while still posing a rigorous test of mathematical reasoning. Information regarding this subset can be found on the Hugging Face Datasets platform (`https://huggingface.co/datasets/HuggingFaceH4/MATH-500`).

- **MAWPS** (Koncel-Kedziorski et al., 2016) is a benchmark focusing on fundamental math word problems (MWPs). It aggregates problems from various sources, primarily involving one or a few arithmetic steps, with a difficulty level roughly corresponding to elementary school mathematics. MAWPS contains 238 test instances and is designed to evaluate model robustness to variations in problem phrasing and numerical values. This dataset is under the MIT License and can be found within the MWPToolkit project at this link.

- **CollegeMath** (Tang et al., 2024) is an emerging dataset designed to assess reasoning abilities on university-level mathematics problems. It comprises 2,818 problems meticulously curated from 9 different university mathematics textbooks, spanning seven core areas such as linear algebra, calculus, probability theory, and differential equations. CollegeMath problems test not only computational skills but also the understanding of advanced mathematical concepts, abstract reasoning, and the application of theorems and methods in complex scenarios, thereby posing a significant challenge to models' generalization and deep reasoning capabilities. This publicly available dataset serves as a valuable resource for evaluating LLMs in the domain of higher education mathematics and is available at this link. For our evaluation, we used a randomly sampled subset of 1,200 items from this dataset.

- **CommonsenseQA** (Talmor et al., 2019) is a multiple-choice question-answering dataset designed to evaluate commonsense reasoning. It contains 12,247 questions, each with 5 options. The model is required to select the most plausible answer, a task that typically cannot be resolved by simple keyword matching but necessitates an understanding of world knowledge, conceptual relationships, and implicit information. CommonsenseQA challenges models' reasoning abilities in non-formal and everyday contexts. The dataset is available at this link and under the CC-BY-4.0 License.

- **MedQA** (Jin et al., 2021) is a professional medical question-answering dataset, with content derived from licensing examinations in the United States (USMLE), mainland China (NM-LEC), and Taiwan (TMQE). Presented primarily as multiple-choice questions, it covers a broad spectrum of medical subfields, including clinical medicine, basic sciences, pharmacology, and diagnostics. MedQA aims to assess models' knowledge mastery, information retrieval capabilities, and, to some extent, clinical reasoning within a highly specialized domain. This dataset is

crucial for advancing research into LLM applications in critical sectors like healthcare. For our experiments, we specifically utilized the official "US" test split from the MedQA dataset. This portion consists of multiple-choice questions designed to evaluate the final test performance on US medical examination questions. The dataset is available at this link and under the MIT License.

# D  MORE DETAILS FOR EXPERIMENT SETTINGS

## D.1  MORE IMPLEMENTATION DETAILS OF COTOOL

The Cognitive Tool-Assisted Reasoning (CoTool) mechanism is integral to processing L4-level complex queries, designed to empower the Large Language Model (LLM) with autonomous external tool invocation capabilities while ensuring operational stability and reliability through robust control measures. Its detailed implementation is as follows:

At each reasoning step within CoTool, the LLM, guided by the detailed **Instruction for CoTool** (as described in App. B.2), first self-assesses whether external tool assistance is required to acquire missing information or perform complex computations. If a tool call is deemed necessary, the LLM generates a specific tool query explicitly stating its needs, which is then encapsulated by special tokens: `<|begin_tool_query|>` and `<|end_tool_query|>`.

Subsequently, after the corresponding tool query is extracted, it is processed by the system. Guided by the **Tool Selection Instruction** (App. B.2), the system accurately selects the most appropriate tool from the RSTKit's (see App. D.2 for details) available suite and constructs the required parameters in **JSON format** for invoking the chosen tool. These parameters, along with the selected tool information, are then passed to RSTKit for execution.

Once the external tool completes its execution, its raw output is returned. The LLM then integrates this information, leveraging the **Task Instruction** (App. B.2). This instruction guides the LLM to synthesize the tool's output with the original user question, the previously generated tool query content, the complete reasoning history up to that point, and the newly acquired tool execution result. The LLM formulates a response that includes an interpretation of the tool's result, how it will be incorporated into the current line of thought, and a concrete plan for the next reasoning step. This synthesized segment is also wrapped with **`<|begin_tool_result|>`** and `<|end_tool_result|>` tokens and seamlessly injected back into the main reasoning chain for subsequent use.

To ensure reliable operation and prevent potential issues such as infinite loops or excessive resource consumption, CoTool incorporates a dual-limiting mechanism:

- **Maximum Tool Calls ($MAX\_TOOL\_CALLS$)**: The system tracks the cumulative number of tool invocations initiated by the LLM during the processing of a single query. If this count reaches the predefined $MAX\_TOOL\_CALLS$ threshold, any subsequent tool call requests are not actually executed. Instead, the LLM receives a specific tool result explicitly indicating that the call limit has been reached (e.g., `reaching max tool call limitations, you cannot use tools anymore`). At this point, the LLM is guided to continue reasoning without relying on external tools or to attempt to summarize the current findings.

- **Maximum Turns ($MAX\_TURN$)**: To control the overall reasoning duration and computational overhead, the system also imposes a $MAX\_TURN$ limit on the entire CoTool reasoning process for a query. A "turn" can be understood as a complete cycle of "LLM deliberation → (optional) tool invocation → LLM integrates the result and continues deliberation." If the number of turns reaches this cap, the reasoning process is forcibly terminated, and the system returns the currently available reasoning results or a status indicating a timeout.

This comprehensive implementation, which combines LLM autonomy in tool use, fine-grained instructional guidance, and strict operational boundaries, enables CoTool to effectively augment the LLM's reasoning capabilities in complex scenarios by leveraging external tools, while simultaneously guaranteeing controllability, stability, and resource efficiency throughout the process.

CoTool operates in a controlled tool environment. The set of available tools, their natural-language descriptions, and their argument formats is fixed and treated as part of the environment dynamics. If

a tool invocation fails (*e.g.*, due to malformed arguments, timeouts, or runtime errors) or returns an invalid result, the corresponding error message is surfaced to the LLM. Under this setting, prompt-injection-like content (*e.g.*, adversarial instructions embedded in user queries or tool outputs) cannot modify the system prompt or the tool definitions, but can only influence intermediate reasoning or tool responses.

### D.2 RSTKIT: REASONING SUPPORT TOOLKIT FOR COTOOL

To effectively handle tool-dependent queries that require external knowledge retrieval or complex computations (defined in 4.1 as Level-4 queries) in *CogER framework*, we develop RSTKit (Reasoning Support Toolkit). RSTKit implements the *Cognitive Tool–Assisted Reasoning (CoTool)* mechanism, providing a suite of standardized external-tool interfaces and unified management features. When an LLM, guided by CoTool, judges its internal knowledge insufficient for a subtask and opts to seek external assistance, it emits a call request to a specific RSTKit tool. An overview of the three primary tool families provided by RSTKit is summarized in Table 9.

RSTKit is designed to provide precise external knowledge access, reliable computation, and flexible code generation capabilities through a powerful, extensible tool-invocation system. It supports various benchmarks, such as GSM8K, MAWPS, CollegeMath, MATH-500, CommonsenseQA, and MedQA, by allowing the LLM to delegate appropriate subtasks.

**Dynamic Tool Registration and Invocation.** During system initialization, all available tools and their metadata (including functionality descriptions, input/output schemas, and sample calls) are auto-registered in a tool registry. At runtime, when the model generates a tool query $q_{\text{tool}}^{(i)}$ (cf. Eq. 9), the system matches $q_{\text{tool}}^{(i)}$ against the registry and selects the most suitable tool. The input arguments are parsed from $q_{\text{tool}}^{(i)}$ and passed in the tool's predefined format; after execution, the tool's output is formatted and returned to the LLM. This loose coupling preserves the independence of the LLM from the implementation of specific tools and improves overall flexibility and maintainability.

**Tool Categories.** RSTKit provides three primary tool families:

- **QA Toolkit.** Empowers the LLM with dynamic access to large-scale external knowledge bases, crucial for queries needing up-to-date or domain-specific background (e.g., CommonsenseQA, MedQA). Core functions include:
  - *Wiki Search:* Given a query string, returns up to a specified number of relevant Wikipedia article titles and page IDs (default top-5), with selectable language.
  - *Page Content Retrieval:* Retrieves the full text of a Wikipedia page by title or ID, in the chosen language.
  - *Page Summary Retrieval:* Retrieves the introductory summary (first few sentences) of a specified Wikipedia article.

- **Multi-functional Calculator Toolkit.** Addresses diverse math-reasoning needs (GSM8K, MATH-500, MAWPS) via three computation modules:
  - *Basic Arithmetic:* Fast, accurate evaluation of +, −, *, /, %, and exponentiation.
  - *Advanced Symbolic Computation:* Leverages SymPy for algebraic simplification, factorization, equation solving, calculus (derivatives, integrals, limits), and matrix operations.
  - *Numerical and Statistical Analysis:* Uses NumPy and SciPy for large-scale array operations, statistical metrics (mean, std, regression), probability distributions, and optimization.

- **Code Generation and Execution Toolkit.** Handles tasks too complex for predefined tools by generating and running custom code:
  - LLM selects code generation and execution tool when it issues a `<|begin_tool_query|>`...`<|end_tool_query|>` tool query (e.g., "Simulate 10 000 coin flips in Python") and determines that other tools cannot solve the problem.
  - The trusted model generates Python code based on information such as tool queries, historical reasoning steps, etc.
  - The code runs in a sandbox with restricted libraries and resources.
  - Outputs (stdout, stderr, return values) are captured, formatted, and returned to the LLM for integration into its reasoning chain.

By integrating these tool families, RSTKit underpins CoTool's automated API invocation within the CogERframework, significantly boosting LLM capability on complex, tool-dependent queries.

Table 9: Overview of RSTKit.

| Tool Category | Sub-category | Num | Description |
|---|---|---|---|
| QA Toolkit | Knowledge Access | 3 | Empowers dynamic access to large-scale external knowledge bases for up-to-date or domain-specific information. |
| Calculator Toolkit | Basic Arithmetic | 8 | Provides fast and accurate evaluation of fundamental arithmetic and mathematical operations. |
| | Symbolic Computation | 13 | Leverages symbolic computation for algebraic manipulation, equation solving, calculus, and matrix operations. |
| | Numerical and Statistical Analysis | 5 | Utilizes numerical libraries for large-scale array operations, statistical analysis, probability calculations, and optimization tasks. |
| Code Generation and Execution Toolkit | Custom Logic Execution | 1 | Handles complex tasks by generating and executing custom Python code. |

### D.3 MORE IMPLEMENTATION DETAILS

We conduct both training and inference processes on NVIDIA $8\times$A800 GPUs, implementing our proposed CogER through the PyTorch[1] framework with version 2.6.0. We train the CogER-Agent for about 1.0 epochs until convergence. To ensure reproducibility, we fix the random seed to 21, 26, and 42, and take the average and standard deviation of three runs. Additionally, we cap all model generations at a maximum of 8192 tokens and accumulate gradients over 4 steps before each optimizer update.

For the hierarchical-aware reward in Eqn. 6, we obtain the minimal sufficient level $L_{\min}(x)$ for each training query $x$ as follows. Before training the CogER-Agent, we evaluate a fixed decoding configuration. For every $x$ in the training set, we run all four predefined reasoning levels $L_1 - L_4$ using the same prompts and decoding hyperparameters as in our experiments, with deterministic decoding (temperature set to 0). We then check whether the final answer at each level matches the ground-truth label and define as $L_{\min}(x) = \min\{\ell \in \{1, 2, 3, 4\} \mid L_\ell \text{ answers } x \text{ correctly}\}$. The estimated $L_{\min}(x)$ is used only during reinforcement learning as part of the hierarchical-aware reward to encode the preference for using the lowest level that is typically sufficient. During training, the CogER-Agent is optimized with GRPO based on the realized accuracy and computational cost of each rollout, and $L_{\min}(x)$ acts as a soft regularizer rather than a hard constraint. At inference time, the agent no longer has access to $L_{\min}(x)$ and selects reasoning levels purely according to the learned routing policy.

---

[1]https://pytorch.org/

# E    CASE STUDY

In this section, we present two case studies illustrating the behavior of our CogER framework when processing L4 queries that necessitate the use of external tools via the CoTool mechanism. L4 queries, as defined in Section 4.1, require creative synthesis, integration of external knowledge, or precise computation beyond the model's internal capabilities. These examples demonstrate how the CogER-Agent, after classifying a query as L4, delegates the task to CoTool, which then autonomously decides when and how to interact with external resources (managed by RSTKit, detailed in App. D.2) to arrive at the solution.

**Case Study 1: CollegeMath - Numerical Evaluation.** This case study originates from the College-Math dataset and involves the numerical evaluation of a polynomial expression. While theoretically solvable by the model through step-by-step arithmetic, such tasks, particularly with potential for computational errors, are appropriately routed as L4 queries to leverage precise external calculation tools provided by RSTKit's Calculator Toolkit.

---

**Case Study 1: CollegeMath**

**Question:** Simplify the expression $n^3 - 9n^2 + 23n - 21$ when $n = 5$.

**Target Answer:** $\boxed{-6}$

- - - - - - - - - - - - - - - - - - - - - - - - - - - - - - - - - - - - - - - - - - - - -

1. To evaluate the expression $n^3 - 9n^2 + 23n - 21$ at $n = 5$, substitute $n = 5$ into the expression:
$$5^3 - 9(5)^2 + 23(5) - 21$$

2. Calculate each term separately:
   - $5^3 = 125$
   - $9(5)^2 = 9 \times 25 = 225$
   - $23 \times 5 = 115$
   - The constant term is $-21$

3. Substitute these values back into the expression:
$$125 - 225 + 115 - 21$$

4. To ensure accuracy, use the calculation tool to compute the entire expression step-by-step:
   **<|begin_tool_query|>**
   $calculate\ 5^3 - 9 * (5)^2 + 23 * 5 - 21$
   **<|end_tool_query|>**

5. **<|begin_tool_result|>**
   - The tool output successfully calculated the expression $5^3 - 9(5)^2 + 23(5) - 21$ and returned the result $-6.00000000000000$. This matches the manual step-by-step calculation I performed earlier (125 - 225 + 115 - 21 = -6), confirming the tool's result is accurate.
   - Next Step: Since the tool's result is correct and the calculation is complete, the final answer can be confidently stated as $-6$.

   **<|end_tool_result|>**
   $\boxed{-6}$

---

**Case Study 2: CommonsenseQA - Knowledge Verification.** This example from the CommonsenseQA dataset requires accessing and verifying specific knowledge about animal habitats. When confronted with a question where its internal knowledge might be insufficient or require confirmation, the model, leveraging CoTool, opts to consult an external knowledge source via RSTKit's QA Toolkit.

---

**Case Study 2: CommonsenseQA**

**Question:** What green area is a marmot likely to be found in?

Please select the most appropriate answer from the following options:

A) countryside   B) great plains   C) encyclopedia   D) jungle   E) north america

**Target Answer:** $\boxed{A}$

- - - - - - - - - - - - - - - - - - - - - - - - - - - - - - - - - - - - - - - - - - - - - - -

1. Marmots are large ground squirrels that typically inhabit specific natural environments.
2. To determine the most appropriate habitat from the given options, I need to recall or verify their typical habitats.
3. The options are: countryside, great plains, encyclopedia, jungle, North America.
4. The encyclopedia (C) is a reference work, not a habitat. Jungle (D) refers to dense forests, which are not typical marmot habitats. North America (E) is a continent, too broad.
5. Marmots are often found in open areas like meadows, grasslands, or mountainous regions.
6. "Countryside" (A) can include such environments, while "great plains" (B) refers to a specific grassland region in North America.
7. To confirm, I need to check the typical habitats of marmots.
   **<|begin_tool_query|>**
      What are the typical habitats of marmots?
   **<|end_tool_query|>**
8. **<|begin_tool_result|>**

   - **Analysis of Tool Output:** The tool `wiki_get_summary` provided a brief summary of marmots, confirming they are ground squirrels found in Asia, Europe, and North America. However, the output does not explicitly mention their habitats (e.g., mountainous regions, meadows, or grasslands). While the summary notes they hibernate underground, this pertains to behavior rather than habitat type. The information is insufficient to confidently answer the question about typical habitats.
   - **Next Step:** Since the initial tool output lacks specific habitat details, I need to answer the query for more deep reasoning.

   **<|end_tool_result|>**
9. The typical habitats of marmots include mountainous regions, alpine meadows, and open grasslands. Among the options provided:

   - Countryside (A) can encompass meadows and open areas where marmots are commonly found.
   - Great Plains (B) refers to a specific grassland region, but marmots are not exclusively tied to this area.
   - North America (E) is too broad, as marmots also inhabit parts of Europe and Asia.

   The most appropriate general answer is A) countryside, as it aligns with the open and varied landscapes marmots inhabit.
   $\boxed{A}$

**Case Study 3: CommonsenseQA - Commonsense Reasoning.** This case study from the CommonsenseQA dataset highlights an inherent system limitation. The problem lacks a valid classification path (L1-L4), meaning any routing decision is destined to fail. The model's assignment to L2 is a symptom of this, as the error was unavoidable regardless of the chosen route.

---

**Case Study 2: CommonsenseQA**

**Question:** What is main benefit to exercising? Please select the most appropriate answer from the following options: A) losing weight B) healthy C) get in shape D) weight loss E) sweat

**Target Answer:** $\boxed{C}$

- - - - - - - - - - - - - - - - - - - - - - - - - - - - - - - - - - - - - - - - - - - - -

To determine the main benefit of exercising, let's analyze each option:
A) Losing weight: While weight loss can be a benefit of exercising, it is not the primary or only benefit. Exercise can help with weight management, but it also has other significant health benefits.
B) Healthy: This option is quite broad and encompasses a wide range of benefits that exercise provides, such as improved cardiovascular health, stronger muscles and bones, better mental health, and more.
C) Get in shape: Getting in shape is a common goal of exercising, but it is more of a general outcome rather than the main benefit. It can be seen as a result of the health improvements that exercise brings.
D) Weight loss: Similar to option A, weight loss is a potential benefit of exercise, but it is not the primary or only benefit.
E) Sweat: Sweating is a physiological response to exercise, but it is not a benefit in itself. It is a byproduct of the body's cooling mechanism during physical activity. Given these considerations, the most comprehensive and accurate answer is:

$\boxed{B}$

---

**Case Study 4: MAWPS - Routing Error leading to Flawed Reasoning.** This case from the MAWPS dataset shows how a routing error causes a reasoning failure. Instead of classifying this arithmetic problem with distractor data as L4 (for calculation) or L3 (for reasoning), the model defaulted to L2. Consequently, its internal reasoning failed to identify and discard the irrelevant information ("6 were torn"). The incorrect final answer is a direct result of this initial misclassification.

---

**Case Study 4: MAWPS**

**Question:** Joan had 695 Pokemon cards , and 6 were torn . Sara bought 133 of Joan 's Pokemon cards . How many Pokemon cards does Joan have now ?

**Target Answer:** $\boxed{562}$

- - - - - - - - - - - - - - - - - - - - - - - - - - - - - - - - - - - - - - - - - - - - -

To determine how many Pokemon cards Joan has now, we need to follow these steps:
1. Start with the initial number of Pokemon cards Joan had. 2. Subtract the number of torn cards. 3. Subtract the number of cards Sara bought from Joan.
Step 1: Joan initially had 695 Pokemon cards.
Step 2: Joan had 6 torn cards, so we subtract these from the initial number:

$$695 - 6 = 689$$

Step 3: Sara bought 133 of Joan's Pokemon cards, so we subtract these from the remaining cards:

$$689 - 133 = 556$$

Thus, the number of Pokemon cards Joan has now is $\boxed{556}$ .

Table 10: Controlled latency analysis of CogER and single-level strategies. "ALL Latency" reports end-to-end latency (s) averaged over all datasets, and "Agent_Latency" reports the additional cost of the CogER-Agent for generating the level tag.

| Version | ID | OOD | Latency | Agent_Latency |
|---|---|---|---|---|
| Oracle | 94.85 | 96.61 | 62.41 | - |
| DeepSeek-R1 | 81.55 | 83.00 | 506.19 | - |
| $L_1$ (Qwen2.5-7B-Instruct) | 76.28 | 86.23 | **50.42** | - |
| $L_2$ (Qwen2.5-32B-Instruct) | 83.62 | 89.49 | 68.52 | - |
| $L_3$ (QWQ-32B) | 86.75 | 93.13 | 147.21 | - |
| $L_4$ (Our CoTool) | 88.42 | 92.89 | 161.22 | - |
| **CogER (Ours)** | **89.28** | **93.56** | 118.53 | **0.01** |

Table 11: Error decomposition of CogER on ID and OOD benchmarks. "Routing Error" denotes cases where the selected level is insufficient, and "Execution Error" denotes cases where the chosen level is sufficient in principle but the underlying reasoning or tool execution still fails.

| Split | Routing Error (%) | Execution Error (%) |
|---|---|---|
| ID | 51.38 | 48.62 |
| OOD | 48.18 | 51.82 |

Table 12: Effect of training set size on CogER performance. The column "Dataset size" denotes the number of training queries used to learn the CogER policy, and "ID" / "OOD" report EM (%) on in-domain and out-of-domain benchmarks, respectively. Bold values indicate the best performance in each column.

| Dataset size | 4K | 6K | **8k** | 10k |
|---|---|---|---|---|
| ID | 86.48 | 86.51 | **89.28** | 86.48 |
| OOD | 92.73 | 92.75 | **93.56** | 92.69 |

## F MORE EXPERIMENTS

**Controlled routing overhead.** To make the efficiency trade-off more explicit, we further isolate the routing cost of the CogER-Agent. As shown in Table 10, the additional latency incurred by the CogER-Agent for generating the level tag (Agent_Latency) is only 0.01 s per query, which is negligible compared with the end-to-end latency of CogER (118.53 s) and with single-level strategies such as $L_1$, whose total latency is 50.42 s. This confirms that the overall efficiency gains of CogER mainly come from avoiding unnecessarily expensive reasoning modes on many queries, while the router itself contributes only a trivial overhead, even for simple $L_1$ cases.

**Quantitative breakdown of failure sources.** To better understand the limitations of CogER, we further decompose its residual errors into decision-related and execution-related failures. As shown in Table 1 and Table 11, CogER achieves 89.28 $EM$ on ID tasks and 93.56 $EM$ on OOD tasks. Among the remaining errors, cases where the CogER-Agent selects an insufficient reasoning level and cases where the downstream reasoning/tool modules fail even at a sufficient level contribute in comparable proportions on both splits (ID: 51.38% *vs.* 48.62%; OOD: 48.18% *vs.* 51.82%). This indicates that the performance is jointly bounded by the level-selection behavior of CogER-Agent and the base reasoning/tool modules rather than being dominated by a single bottleneck.

**Impact of training set size.** We investigate how the amount of supervision affects CogER by varying the training set size while keeping the data mixture and all other settings fixed. From Table 12, CogER already achieves competitive performance with 4K training samples, and the results remain relatively stable as the dataset grows. The 8K configuration yields the best $EM$ on both ID and OOD tasks, suggesting that CogER is effectively trained with a moderate number of examples.

**Accuracy within routed levels.** We examine how well each reasoning level performs on the subset of queries it actually handles. As shown in Table 13, the accuracy within each routed subset remains high on both ID and OOD tasks, with most levels achieving over

Table 13: Accuracy (%) of queries routed by the CogER-Agent to each reasoning level on ID and OOD tasks. Acc($L_i$) denotes the $EM$ accuracy computed over the subset of queries that are dynamically routed to level $L_i$.

| Version | ID | OOD |
|---|---|---|
| Acc($L_1$) | 95.45 | 100.00 |
| Acc($L_2$) | 92.45 | 97.20 |
| Acc($L_3$) | 95.82 | 95.06 |
| Acc($L_4$) | 92.60 | 90.64 |

90% $EM$. This indicates that, conditioned on the level selected by CogER, each reasoning mode is generally reliable on the queries it receives.

**Impact of Agent size.** We study how the capacity of the CogER-Agent affects performance by varying its size while keeping all expert models and training settings fixed. From Table 14, all three variants with 3B, 7B, and 14B agents achieve strong results on MAWPS (97.06%, 97.87%, and 98.32% $EM$, respectively). Scaling the agent from 3B to 7B brings a small improvement, and enlarging it further to 14B yields only a marginal additional gain at the cost of higher computation. These results indicate that CogER is robust to the choice of agent size and that the routing problem does not require an excessively large model.

Table 14: Comparison of different agent sizes on MAWPS.

| Agent Size | $EM$ |
|---|---|
| Qwen2.5-3B-Instruct | 97.06 |
| Qwen2.5-7B-Instruct | 97.87 |
| Qwen2.5-14B-Instruct | **98.32** |

**Per-level routing behavior.** We analyze the routing decisions by reporting per-level precision, recall, and F1-score. From Table 15, we observe that $L_1$ and $L_2$ achieve very high precision (97.53% and 96.81%), indicating that when CogER predicts a low reasoning level, this choice is usually appropriate for the query. Their relatively lower recall shows that many harder queries are correctly escalated to higher levels instead of being over-confidently kept

Table 15: Per-level routing precision(%), recall(%), and F1-score(%) of the CogER-Agent on all benchmarks.

| Version | Precision | Recall | F1-score |
|---|---|---|---|
| $L_1$ | 97.53 | 39.27 | 55.99 |
| $L_2$ | 96.81 | 98.68 | 97.73 |
| $L_3$ | 20.80 | 85.09 | 33.43 |
| $L_4$ | 36.67 | 4.59 | 8.15 |

at $L_1/L_2$. In contrast, $L_3$ and $L_4$ exhibit much higher recall but lower precision, which is consistent with their role as "catch-all" options for more difficult or ambiguous problems: most truly hard queries are routed to these levels, while some borderline cases are conservatively upgraded as well. Together with the overall EM results, this analysis suggests that CogER learns a reasonable and interpretable routing pattern rather than relying on a single level, and that residual errors are jointly influenced by both level selection and the underlying reasoning/tool modules.

# G   Discussions and Future Works

## G.1   Limitations and Future Works

In this paper, we propose **Cog**nition-**E**lastic **R**easoning (**CogER**), a four-tier cognitive hierarchy inspired by human layered reasoning mechanisms, which dynamically selects the most appropriate processing mode for each query. However, we believe that there are potential studies worth exploring in the future to further capitalize on the advantages of CogER:

- **Extension to Interactive and Multi-Modal Settings.** CogER has been validated on single-turn, text-only tasks. Its performance in conversational or multi-modal contexts (*e.g.*, image-based reasoning) remains untested. In the future, we will extend the framework to maintain and update per-query context across multiple turns and to incorporate vision and other modalities into complexity estimation and reasoning tiers, thereby enabling truly general adaptive inference.

- **Reward Sparsity and Alignment.** The current reward function may be too coarse to capture the nuanced quality of complex reasoning. In long CoT or creative tasks, these sparse signals can lead to unstable policy learning or unintended "reward hacking." In the future, we will integrate richer supervisory signals such as human preference feedback, apply inverse reinforcement learning to infer underlying reward structures, and develop multi-objective reward optimization with adaptive weight tuning to ensure that reward signals robustly align with real-world task requirements.

- **Agent-Execution Co-design.** Our error analysis shows that residual failures are roughly evenly split between cases where the CogER-Agent selects an insufficient level and cases where the downstream reasoning/tool modules still fail even at a sufficient level. This suggests that future work should not only focus on more expressive level-selection policies of the agent, but also on more robust reasoning and tool modules, as well as joint training schemes that explicitly coordinate the agent's decisions with the execution components.

## G.2   Broader Impacts

**Positive Societal Impacts.** By dynamically allocating inference effort per query, our CogER framework substantially reduces average compute, which can translate into lower energy consumption and carbon emissions for large-scale deployments. Moreover, by enabling on-demand invocation of specialized external tools, our approach can improve reliability and factual grounding in critical applications, *e.g.*, medical question answering, scientific data analysis, and legal research, thus enhancing trust and enabling broader societal benefit from AI.

**Negative Societal Impacts.** As with any powerful AI technology, there is a risk that our method could be used for malicious purposes. For example, to generate convincing fake content.

## Use of Large Language Models Disclosure

In accordance with the ICLR 2026 policy on LLM usage, we disclose that our study did not use any LLM to generate scientific content or perform major experiments. The only use of an LLM (ChatGPT-5) was to polish the English writing and improve presentation quality; all core methodology, experiments, and analyses were authored and verified by the human authors.

