# OpenReview forum: "Beyond Fast and Slow: Cognitive-Inspired Elastic Reasoning for Large Language Models"
_ICLR.cc/2026/Conference — ICLR 2026 Conference Desk Rejected Submission_

### Official Review · Reviewer_Bhkw · 2025-10-29

**Soundness:** 3
**Presentation:** 2
**Contribution:** 2
**Rating:** 4
**Confidence:** 4

**Summary:**

This paper proposes a dynamic reasoning framework called CogER (Cognitive-Inspired Elastic Reasoning), which aims to address the inefficiency in LLM reasoning caused by the one-size-fits-all strategy. The paper divides query tasks into four levels (L1-L4) based on cognitive complexity. Subsequently, the paper uses RL to train a "CogER-Agent." For effective training, the paper designs a composite reward function that includes a "Hierarchical-Aware Reward," which penalizes "overthinking" (i.e., using a more complex strategy than the required level). The experimental results show that CogER significantly reduces latency while achieving higher accuracy.

**Strengths:**

1. The authors propose a novel classification perspective and a new method for understanding problem difficulty.

2. Experiments were conducted on multiple In-domain and Out-of-Domain datasets, and detailed ablation studies were also included.

**Weaknesses:**

1. Regarding "the minimal level required for a given query," the paper does not provide a reasonable explanation. How is this obtained? If the question-solving ability is not stable (e.g., L2 might occasionally be able to solve it correctly), how to handle this? I believe this is a crucial point of the paper, but it is not discussed.

2. There is confusion between the cognitive hierarchy and tool requirements. Problem complexity and the need for tools are two orthogonal attributes. A very complex mathematical calculation (e.g., 9-digit multiplication) might be solvable with "L1+tool," so I do not fully understand the rationale for L4 and designing the CoTool component. Its contribution to the main theme does not seem sufficient.

**Questions:**

See weaknesses.

---

> ### Author Response · Authors · 2025-11-22
> **Response to Reviewer Bhkw (Part 1)**
>
> We sincerely thank you for your thoughtful review and for recognizing the strengths of our work, particularly our **novel classification perspective and new method** for understanding problem difficulty. We also greatly appreciate your positive assessment of our comprehensive experimentation on both in-domain and out-of-domain datasets, as well as the **detailed ablation studies** included in the paper.
>
> >Q1. Regarding "the minimal level required for a given query," the paper does not provide a reasonable explanation. How is this obtained? If the question-solving ability is not stable (e.g., L2 might occasionally be able to solve it correctly), how to handle this? I believe this is a crucial point of the paper, but it is not discussed.
>
> **A1.** We thank the reviewer for raising this important question and agree that the notion of “the minimal level required for a given query” should be made more explicit. We clarify this in the revised version (**see in Appendix D.3**).
> * **How we obtain the minimal level $L_{\min}(x)$.** For each training query $x$, we empirically define $L_{\min}(x)$ as the **smallest reasoning level among $\{L1, L2, L3, L4\}$ that can correctly solve the query under a fixed decoding setup**. Specifically, before training the CogER-Agent, we run all four levels on each training query (using the same prompts and decoding hyper-parameters as in the main experiments) and check whether the final answer is correct. We then set $L_{\min}(x) = \min \{\ell \in {1,2,3,4} \mid L_\ell \text{ answers } x \text{ correctly}\}$. Importantly, this oracle-like $L_{\min}(x)$ is used **only during RL training** as part of the Hierarchical-Aware Reward. At test time, CogER does *not* require access to $L_{\min}$.
> * **How we handle unstable solving ability.** We do not assume that $L_{\min}(x)$ is a perfectly accurate or deterministic label. Instead, it is used as a **soft regularizer** encoding the preference “use the simplest level that is usually sufficient.” The policy is actually learned from the realized accuracy and computational cost of each rollout. We optimize the CogER-Agent with group-relative policy optimization (GRPO), where rewards are normalized within a group of rollouts for the same query. Therefore, if $L2$ can reliably solve some queries that were initially labeled with $L_{\min}=L3$, the successful $L2$ trajectories will obtain **higher overall returns** (higher accuracy with lower cost), and the policy will still learn to favor $L2$ on such queries despite the mild hierarchical penalty. Conversely, when lower levels are unstable and often fail, their lower expected accuracy naturally discourages the agent from selecting them, even if they were sometimes assigned as $L_{\min}$.

---

> ### Author Response · Authors · 2025-11-22
> **Response to Reviewer Bhkw (Part 2)**
>
> >Q2. There is confusion between the cognitive hierarchy and tool requirements. Problem complexity and the need for tools are two orthogonal attributes. A very complex mathematical calculation (e.g., 9-digit multiplication) might be solvable with "L1+tool," so I do not fully understand the rationale for L4 and designing the CoTool component. Its contribution to the main theme does not seem sufficient.
>
> **A2.** We thank the reviewer for the thoughtful comment. We address this concern from two perspectives: (1) how L1–L4 are defined in this work, and (2) how the CoTool component fits into the overall theme of elastic reasoning.
> * **Rationale for L4 and how the hierarchy is defined.** In our paper, inspired by Bloom’s Taxonomy, the four **complexity levels** L1–L4 are **operationalized as different reasoning modes** (Sec. 4.1):
>     * **L1 – Prompt Answering**: direct responses without explicit reasoning.
>     * **L2 – CoT Reasoning**: short chain-of-thought for basic reasoning and comprehension.
>     * **L3 – Deep Reasoning**: longer multi-step reasoning for more involved analysis and evaluation.
>     * **L4 – Tool-Enhanced Deep Reasoning**: deep reasoning combined with explicit tool calls for queries that benefit from precise computation or external knowledge.
>
>     Under this task-oriented definition, **we deliberately group tool-dependent queries into L4 and treat it as the highest level in our hierarchy**. For the reviewer’s example of 9-digit multiplication, our design simply regards such cases as L4-type queries: in practice, the learned L4 behavior often looks like “a very short internal reasoning step + one tool call,” which is functionally similar to an “L1+tool” pattern. Instead of introducing a separate “L1+tool” level, we subsume these behaviors into L4 to keep the hierarchy compact and conceptually clear: **L1–L3 cover purely internal reasoning with increasing depth, while L4 represents a unified tool-augmented reasoning mode built on top of this hierarchy.**
> * **Necessity and contribution of the CoTool component.** The CoTool component is essential because it is the concrete mechanism that realizes L4 and makes CogER elastic **over both internal computation and external assistance**:
>     * **Without CoTool and L4**, CogER would reduce to choosing among L1–L3 only, i.e., adjusting how much the model thinks internally, similar in spirit to existing test-time scaling methods.
>     * **With CoTool**, L4 becomes a distinct action where the CogER-Agent can switch from internal reasoning (L2/L3) to tool-augmented reasoning when this yields a better accuracy–cost trade-off (e.g., for complex arithmetic or knowledge-intensive queries). In this way, CogER learns not only which internal level to use (L1/L2/L3), but also when to escalate to the tool-enhanced L4 mode for harder queries.
>
>     Empirically, Table 5 shows that enabling CoTool improves EM on MATH-500 and CollegeMath while tools are invoked on only a small fraction of queries. These results indicate that L4 and CoTool provide a necessary elastic option in our framework rather than a marginal add-on: they allow CogER to dynamically decide both “how deeply to reason internally” and “when external tools should be brought in,” which directly supports the central goal of Cognitive-Inspired Elastic Reasoning.
>
> ***
> We sincerely hope our clarifications above have addressed your concerns. We would be grateful if you could kindly reconsider the evaluation of our paper.

---

> ### Author Response · Authors · 2025-11-25
> **Looking forward to the response from Reviewer Bhkw**
>
> Dear Reviewer Bhkw,
>
> We have tried our best to address all the concerns and provided explanations for all the questions. We sincerely hope that our answer has addressed your initial concerns. Kindly let us know if you have any other concerns, and we will do our best to address them.
>
> Best regards,
>
> The Authors

---

> ### Author Response · Authors · 2025-11-28
>
> Dear Reviewer Bhkw,
>
> Thank you for your constructive feedback. We also appreciate your initial assessment, particularly your comments highlighting "... propose **a novel classification perspective** and **a new method** for understanding problem difficulty" and "**Experiments were conducted on multiple** In-domain and Out-of-Domain **datasets**, and **detailed ablation studies** were also included".
>
> We would like to take this opportunity to reiterate the core contributions of our work, which we believe make a significant contribution to the field:
> * **Cognitive-inspired elastic reasoning framework.** We propose **CogER**, a four-level cognitive hierarchy that dynamically selects among reasoning modes at test time, enabling flexible control of reasoning depth and computational cost.
> * **Hierarchical reward for accuracy–efficiency trade-offs.** We design a composite reward that combines format correctness, answer accuracy, and a hierarchy-aware term based on the minimal sufficient level, explicitly discouraging overthinking while maintaining reliability.
> * **Tool-augmented deep reasoning with CoTool.** We introduce **CoTool** and the accompanying RSTKit toolkit to integrate external tools into the highest reasoning level, allowing the model to decide when to invoke tools and to incorporate tool outputs into its reasoning process.
> * **Extensive empirical validation.** Experiments on ID and OOD reasoning benchmarks show that CogER achieves stronger accuracy–efficiency trade-offs than fixed-depth and routing-based baselines.
>
> We hope that these innovations and the substantial empirical gains will merit a score improvement. Should you have any further questions, we are happy to provide additional clarifications.
>
> Thank you again for your time and consideration.
>
> Sincerely,
>
> The Authors

---

### Official Review · Reviewer_Y9xi · 2025-10-30

**Soundness:** 3
**Presentation:** 2
**Contribution:** 3
**Rating:** 4
**Confidence:** 3

**Summary:**

This paper proposes CogER, a framework to address the inefficiency of uniform LLM reasoning. It dynamically allocates computation by classifying queries into four complexity levels (L1-L4), inspired by cognitive science. An RL-trained "CogER-Agent" routes queries to one of four corresponding strategies. A composite reward function is designed to balance accuracy and computational cost. Experiments show CogER achieves SOTA accuracy on various reasoning benchmarks while dramatically reducing end-to-end latency compared to strong baselines.

**Strengths:**

1.  Clear Motivation & Principled Design: The paper clearly targets the "one-size-fits-all" inefficiency. The 4-level hierarchy derived from cognitive science (Bloom's Taxonomy) provides a logical and well-founded structure.
2.  Strong Experimental Results: CogER achieves SOTA accuracy on both ID and OOD tasks. The efficiency gains are significant, demonstrating, for example, over 4x lower latency than the top-performing DeepSeek-R1 baseline.
3.  Effective Reward Function: The composite reward is well-designed. The $\mathcal{R}_{hierarchy}$ component, which penalizes "overthinking," is proven critical by the ablation study (Table 4) for preventing the agent from defaulting to the most expensive strategy.

**Weaknesses:**

1.  Lack of Controlled Routing Overhead Analysis: The paper does not quantify the specific latency overhead in the controlled environment. This makes it difficult to ascertain the precise efficiency trade-off, especially for simple L1 queries where the router's cost may be non-trivial.

2.  Absence of Error Analysis: There is no breakdown analysis of the framework's failure cases. It is unclear whether errors stem from (1) the agent's incorrect routing or (2) the execution module's failure despite correct routing. This analysis is needed to understand the model's limitations.

**Questions:**

1.  What is the CogER-Agent's "routing accuracy" in terms of predicting the minimal sufficient level ($L_{min}$)? The classifier might be wrong. Relatedly, what is the "oracle" accuracy of the CogER framework, assuming a 100% perfect router?

2.  Could the authors provide the accuracy within each dynamically routed level? (i.e., for the set of all queries the agent routed to L1, what was the accuracy? And similarly for L2, L3, and L4?)

---

> ### Author Response · Authors · 2025-11-22
> **Response to Reviewer Y9xi (Part 1)**
>
> We are deeply grateful to you for recognizing the strengths of our work, particularly the **clear motivation and principled four-level hierarchy grounded in Bloom’s Taxonomy**, the strong experimental results where CogER achieves **SOTA accuracy and substantial latency reductions**, and the effectiveness of our composite reward.
>
>
> >Q1. Lack of Controlled Routing Overhead Analysis: The paper does not quantify the specific latency overhead in the controlled environment. This makes it difficult to ascertain the precise efficiency trade-off, especially for simple L1 queries where the router's cost may be non-trivial.
>
> **A1.** We thank the reviewer for pointing out the need for a controlled analysis of routing overhead. In the revised version (**see in Appendix F**), we add a dedicated latency breakdown that quantifies the cost of the CogER-Agent itself and clarifies the efficiency trade-off, including for simple L1 queries.
> * **Controlled routing overhead is negligible.** We report a controlled routing overhead analysis in Table A. The results show that the Agent_Latency is only **0.01 s per query**, which **is negligible compared with the overall CogER latency** of 118.53 s and with the single-level strategies such as $L_1$ whose total latency is 50.42 s.
> * **Overall efficiency trade-off remains favorable.** These results confirm that the efficiency trade-off remains favorable even after explicitly accounting for routing overhead. CogER still achieves a much lower latency than strong baselines such as DeepSeek-R1 (118.53 s vs. 506.19 s), while obtaining higher accuracy on both ID and OOD benchmarks. In other words, the small additional cost of the CogER-Agent is far outweighed by the savings from avoiding unnecessarily expensive reasoning modes on many queries.
> * **Router cost for L1 queries is trivial.** Regarding simple L1 queries, Table A indicates that the router’s cost is also trivial in this regime. For queries that are routed to $L_1$, the additional 0.01 s spent by the CogER-Agent on generating the level tag is extremely small relative to the full processing time of the L1 strategy itself.
>
> Table A: Controlled latency analysis of CogER and single-level strategies. “ALL Latency” reports end-to-end latency (s) averaged over all datasets, and “Agent\_Latency” reports the additional cost of the CogER-Agent for generating the level tag.
> |Version| ID | OOD |ALL Latency|Agent_Latency|
> |-|-|-|-|-|
> |DeepSeek-R1|81.55|83.00|506.19|-|
> |$L_1$ (Qwen2.5-7B-Instruct)|76.28 | 86.23|**50.42**|-|
> |$L_2$ (Qwen2.5-32B-Instruct) | 83.62 | 89.49|68.52|-|
> |$L_3$ (QWQ-32B)| 86.75| 93.13|147.21|-|
> |$L_4$ (Our CoTool)| 88.42|92.89|161.22|-|
> |**CogER (Ours)** |**89.28**|**93.56**|118.53|**0.01**|
>
> >Q2. Absence of Error Analysis: There is no breakdown analysis of the framework's failure cases. It is unclear whether errors stem from (1) the agent's incorrect routing or (2) the execution module's failure despite correct routing. This analysis is needed to understand the model's limitations.
>
> **A2.** We thank the reviewer for pointing out the need for a more detailed error analysis. We added a breakdown of failure cases to distinguish routing errors from execution errors and to better expose the limitations of CogER.
> * **Quantitative breakdown of failure sources.** CogER achieves 89.28% EM on ID tasks and 93.56% EM on OOD tasks. Among the remaining errors, on ID tasks, 51.38% of failures are due to incorrect routing (the agent selects an insufficient level), and 48.62% are due to execution errors given a correct or more conservative level. On OOD tasks, 48.18% of failures are attributed to routing and 51.82% to execution. We will report these numbers in the revised version (Appendix F) to make the behavior of CogER more transparent.
> * **Implications for limitations and future work.** This analysis shows that both components contribute to the residual error. This highlights two main limitations of the current framework: the router still has room for improvement in fine-grained level selection, and the execution modules at L3/L4 can struggle with very challenging problems. We will emphasize these observations and point out that strengthening both the routing policy and the base reasoning/tool modules constitutes an important direction for future work.

---

> ### Author Response · Authors · 2025-11-22
> **Response to Reviewer Y9xi (Part 2)**
>
> >Q3. What is the CogER-Agent's "routing accuracy" in terms of predicting the minimal sufficient level ($L_{min}$)? The classifier might be wrong. Relatedly, what is the "oracle" accuracy of the CogER framework, assuming a 100% perfect router?
>
> **A3.** We thank the reviewer for this constructive question. We clarify both the role of $L_{\min}(S)$ and the routing performance of CogER, including an oracle upper bound.
> * **Role of $L_{\min}(S)$ and routing objective.** In our framework $L_{\min}(S)$ is used in the Hierarchical-Aware Reward $R_{\text{hierarchy}}$ (Eq. (6)–(8)) as a preference signal that **encourages the agent to avoid unnecessarily high levels when they are not needed**. The policy is still optimized with respect to the combined reward $R_{\text{format}} +R_{\text{accuracy}} + R_{\text{hierarchy}}$. In other words, the **training objective is to maximize answer accuracy while reducing computational cost**, rather than to exactly classify each query to its estimated $L_{\min}(S)$. Therefore, we do not expect the router to perfectly match $L_{\min}(S)$ on every example, and slight deviations toward stronger levels are acceptable when they improve reliability. Under this setting, the strict top-1 routing accuracy with respect to $L_{\min}(S)$ is 38.88%, which is expected because the policy is not trained as a pure classifier of $L_{\min}(S)$, while the **task-sufficient routing accuracy**, defined as routing to a level that successfully solves the query, **reaches 93.93%**.
> * **Oracle performance with perfect knowledge of $L_{min}(S)$**. To address the “oracle accuracy” question, we add an oracle policy in the **Table B**. This oracle is given perfect knowledge of $L_{\min}(S)$ for every query and always selects that level, which corresponds to a router with 100% accuracy under our estimated minimal-level labels. As shown in Table B, the oracle achieves 94.85% EM on ID tasks and 96.61% on OOD tasks, with an average latency of 62.41 s. This serves as a *theoretical upper bound* that relies on privileged information not available in real deployments. In contrast, CogER does **not** observe $L_{\min}(S)$ at test time, yet it still attains 89.28% ID and 93.56% OOD with a latency of 118.53 s. We will explicitly present this oracle as an upper bound in the paper and position **CogER as a practical, learnable routing policy that approaches this ideal performance without requiring oracle access to $L_{\min}(S)$**.
>
>
> Table B: Accuracy (\%) of each reasoning mode and the CogER on ID and OOD tasks.
>
> |Version| ID | OOD |Param.$\downarrow$ |Latency$\downarrow$ |Words$\downarrow$ |
> |-|-|-|-|-|-|
> |**Oracle**|94.85|96.61|12.2B|62.41|304.31|
> |$L_1$ (Qwen2.5-7B-Instruct)|76.28 | 86.23|7B|50.42|817.71|
> |$L_2$ (Qwen2.5-32B-Instruct) | 83.62 | 89.49|32B|68.52|781.20|
> |$L_3$ (QWQ-32B)| 86.75| 93.13|32B|147.21|774.75|
> |$L_4$ (Our CoTool)| 88.42|92.89|32B|161.22|895.51|
> |CogER (Ours)|89.28|93.56|29.6B|118.53|489.71|
>
> >Q4. Could the authors provide the accuracy within each dynamically routed level? (i.e., for the set of all queries the agent routed to L1, what was the accuracy? And similarly for L2, L3, and L4?)
>
> **A4.** We thank the reviewer for this helpful suggestion. We provide the requested accuracy statistics for each reasoning level under the actual routing decisions of CogER.
> * **Accuracy for queries routed to each level.** For every test query, we record the level chosen by the CogER-Agent and then compute the exact-match accuracy within the subset of queries routed to that level. Denoting these quantities as $\text{Acc}(L_i)$, Table C reports the results on both ID and OOD benchmarks. We observe that the accuracy within each routed subset is consistently high. These results indicate that, conditioned on the router’s decision, **each reasoning mode is applied to a subset of queries on which it is mostly successful.**
> * **Implication for routing quality.** The above analysis shows that errors are not dominated by a single “bad” level that receives many unsolved queries. Instead, all four levels achieve strong accuracy on the queries they handle. This supports our claim that **CogER learns a reasonable routing policy**, where L1 mainly receives easier questions and higher levels L2–L4 focus on progressively harder ones, while still solving the majority of routed queries.
>
> Table C:  Accuracy (%) of queries routed by the CogER-Agent to each reasoning level on ID and OOD benchmarks. $\text{Acc}(L_i)$ denotes the exact-match accuracy computed over the subset of queries that are dynamically routed to level $L_i$.
>
> |Version|ID|OOD|
> |-|-|-|
> |$\text{Acc}(L_1)$|95.45|100.00|
> |$\text{Acc}(L_2)$|92.45|97.20|
> |$\text{Acc}(L_3)$|95.82|95.06|
> |$\text{Acc}(L_4)$|92.60|90.64|
>
> ***
> We sincerely hope our clarifications above have addressed your concerns. We would be grateful if you could kindly reconsider the evaluation of our paper.

---

> ### Author Response · Authors · 2025-11-25
> **Looking forward to the response from Reviewer Y9xi**
>
> Dear Reviewer Y9xi,
>
> We have addressed your initial concerns regarding our paper. We are happy to discuss them with you in the OpenReview system if you feel that there are still some concerns/questions. We also welcome new suggestions/comments from you!
>
> Best regards,
>
> The Authors

---

> ### Comment · Reviewer_Y9xi · 2025-11-28
>
> I appreciate the comprehensive feedback and the new experimental results.
> The explanation provided in A3 is very insightful, and helped me to understand the difference from the theoretical bound.
> I have no further questions at this stage and plan to raise my score.
>
> Please note that I cannot edit the score at this moment due to a system issue on openreview. I will proceed with the update as soon as possible.

---

> > ### Author Response · Authors · 2025-11-28
> >
> > Dear Reviewer Y9xi,
> >
> > Thank you for your constructive feedback. We are delighted that you found our responses satisfactory.
> >
> > Should you have any further questions, we are happy to provide additional clarifications.
> >
> > Thank you again for your time and consideration.
> >
> > Sincerely,
> >
> > The Authors

---

### Official Review · Reviewer_EgK3 · 2025-10-31

**Soundness:** 2
**Presentation:** 3
**Contribution:** 3
**Rating:** 4
**Confidence:** 2

**Summary:**

This paper addresses inefficiencies in using fixed reasoning strategies in LLMs. It introduces CogER, a cognitive-inspired framework that dynamically matches query complexity with adaptive reasoning modes, ranging from direct answers to tool-assisted reasoning. Experimental results on various datasets demonstrate that CogER improves accuracy, reduces latency, and decreases token generation compared to fixed strategies and scaling-based baselines.

**Strengths:**

- The paper proposes categorizing queries by complexity into different reasoning modes, including direct answer, concise CoT, extended CoT, and tool-assisted reasoning. This method provides a novel mechanism to balance accuracy and computational efficiency.

- The approach demonstrates significant improvements across multiple benchmarks, achieving notable gains in accuracy, efficiency, and reduced latency compared to standard fixed or scaling-based strategies.

- By drawing inspiration from cognitive models, the paper introduces hierarchical reasoning strategies that align with human reasoning complexity. This innovative approach improves interpretability and realism of model reasoning.

**Weaknesses:**

- The reward depends on $L_{\min}(\mathcal{S})$, the minimal sufficient level, but the paper does not explain how this unobservable quantity is obtained during training or evaluation.

- It is not clear how to handle tool errors and prompt injection, and how to avoid gaming of the format reward by printing tags without real gains.

- The MDP action space mixes high-level actions with the token vocabulary $\mathcal{V}$. It would be beneficial if the authors could further explain how actions are masked or factorised.

**Questions:**

Please check the weaknesses.

---

> ### Author Response · Authors · 2025-11-22
> **Response to Reviewer EgK3 (Part 1)**
>
> We are deeply grateful to you for recognizing the strengths of our work, particularly that **CogER provides a novel mechanism** to balance accuracy and computational efficiency, achieving **notable gains in accuracy, efficiency, and reduced latency**.  We also sincerely appreciate your positive comments on the cognitively inspired hierarchical strategies.
>
> >Q1. The reward depends on $L_{min}(S)$, the minimal sufficient level, but the paper does not explain how this unobservable quantity is obtained during training or evaluation.
>
> **A1.** We thank the reviewer for highlighting this important point. In the revised version, we added a detailed description of how we estimate $L_{min}(S)$ in **Appendix D.3.** We clarify the role of $L_{min}(S)$ from two aspects:
> * **Estimation of $L_{min}(S)$ during training.**
>    * In practice, $L_{min}(S)$ is **estimated on the training set** using the underlying model under fixed strategies.
>     * For each training query $S$, we run the LLMs once with each fixed level $L_1, L_2, L_3, L_4$ (corresponding to our four reasoning modes) using the same prompts and a deterministic decoding configuration (e.g., temperature set to $0$), and record whether the final answer is correct.
>    * We then define $L_{min}(S)$ as the smallest level among $\{L_1,L_2,L_3,L_4\}$ that successfully solves the query. If several levels succeed, we pick the lowest one.
>    * This yields a model-aware, task-specific approximation of the “minimal sufficient level” for each training query, which is exactly what is used in the hierarchical-aware reward in Eq. (6)–(8).
> * **Usage of $L_{min}(S)$ during evaluation.**
>    * At test time, the CogER-Agent selects a reasoning level according to its learned policy, executes the corresponding strategy, and we report standard metrics (EM, latency, token usage) on the test sets.
>    * $L_{min}(S)$ appears **only in the training reward as a shaping signal** and is never assumed to be observable at test time, so the evaluation protocol remains fully realistic and comparable to existing methods.
>
>
> >Q2. It is not clear how to handle tool errors and prompt injection, and how to avoid gaming of the format reward by printing tags without real gains.
>
> **A2.** We thank the reviewer for raising these important practical issues. In our current setup, tool errors and potential prompt injection are discouraged through the accuracy driven reward and the design of a controlled tool interface, and the format reward plays a foundational gate role that still cannot be exploited by printing tags alone.
>
> * **Handling tool errors and prompt injection in our setting.**
>     * In our framework, tool calls within CoTool (for example, a calculator or external knowledge access) are treated as part of the environment dynamics of the L4 reasoning mode. If a tool call fails (for example, due to malformed arguments or a runtime error) or returns an incorrect intermediate result, the final answer is typically wrong and therefore receives no accuracy reward $R_{\text{accuracy}}$. As a result, policies that overuse tools or produce unreliable tool calls are naturally disfavored during GRPO optimization because they combine low accuracy with higher cost.
>     * For prompt injection, our experiments are conducted under a controlled tool interface, where the available tools, their descriptions, and their argument formats are fixed, and tools cannot modify the reward computation or the training process. Under this setting, prompt-injection-like content (for example, adversarial instructions that attempt to override the original prompt) can at most distort intermediate tool outputs or internal reasoning. We added a clarification of this controlled tool environment and its robustness assumptions in **Appendix D.1 of the revised manuscript**.
> * **Avoiding gaming of the format reward by printing tags.**
>    * First, $R_{\text{format}}$ only checks structural correctness and self-consistency between the declared level tag and the executed reasoning pattern. For example, L2 requires a concise chain of thought, and L4 requires at least one tool call. It does not depend on the content of the final answer.
>    * Second, even if the model prints a syntactically correct tag, an incorrect final answer still yields $R_{\text{accuracy}}(S,A) = 0$, and unnecessarily choosing high levels can incur a negative contribution in $R_{\text{hierarchy}}(S,A)$. Consequently, trajectories that merely print tags without actually solving the problem obtain a much smaller total return than trajectories that both respect the format and produce correct answers at appropriate levels. In our GRPO training, we did not observe policies that exploit format-only behaviors.

---

> ### Author Response · Authors · 2025-11-22
> **Response to Reviewer EgK3 (Part 2)**
>
> >Q3. The MDP action space mixes high-level actions with the token vocabulary $\mathcal{V}$. It would be beneficial if the authors could further explain how actions are masked or factorised.
>
> **A3.**  We thank the reviewer for this insightful comment. Our MDP definition in Sec. 4.2 is intended to describe the full environment, while in implementation the effective decision space is factorised around the four complexity levels $L_1–L_4$. We clarify this as follows:
> * **Role of high-level actions and the vocabulary in our formulation.**
>    * In Sec. 4.2 we write the action space as $A = \{\text{No-Think}, \text{Think}, \text{Extend}, \text{Delegate}, \mathcal{V}\}$ to conceptually capture that the agent both (i) chooses a **reasoning mode** (No-Think/Think/Extend/Delegate, corresponding to $L_1–L_4$ in Sec. 4.1), and (ii) produces token-level outputs from the vocabulary $\mathcal{V}$.
>    * The 7B CogER-Agent is guided by the **system prompt in Appendix B.1** to first classify the query into one of the four levels and encode this decision using the special tags `<question_level>Li</question_level>`. This level tag is the concrete realisation of the high-level actions in the MDP, while subsequent reasoning for each level follows the rollout description in Sec. 4.2 (L1: direct answer; L2: concise CoT; L3: extended CoT; L4: Delegate via CoTool).
> * **Factorised decision process: level selection followed by level-specific processing.**
>    * During rollout, the system prompt explicitly instructs the agent to output the level tag first. At this stage, the effective action space is reduced to the small set of level tokens that correspond to $L_1–L_4$, other vocabulary tokens are discouraged by the format requirement and the reward in Sec. 4.3. This can be viewed as a **soft masking** of $\mathcal{V}$ down to the four high-level choices.
>    * Once the level has been identified: **1) L1 (No-Think)**: the agent immediately continues to generate the final answer itself without additional reasoning. **2) L2–L4 (Think/Extend/Delegate)**: the predicted level is read from `<question_level>Li</question_level>`, and the query is passed to the corresponding reasoning module (concise CoT, extended CoT, or CoTool) as described in Sec. 4.4. The token generation inside these level-specific modules operates over $\mathcal{V}$ but is treated as part of the environment dynamics rather than as separate RL actions of the CogER-Agent.
>
> ***
> We sincerely hope our clarifications above have addressed your concerns. We would be grateful if you could kindly reconsider the evaluation of our paper.

---

> ### Author Response · Authors · 2025-11-25
> **Looking forward to the response from Reviewer EgK3**
>
> Dear Reviewer EgK3,
>
> We have tried our best to address all the concerns and provided explanations for all the questions. We sincerely hope that our answer has addressed your initial concerns. Kindly let us know if you have any other concerns, and we will do our best to address them.
>
> Best regards,
>
> The Authors

---

> ### Author Response · Authors · 2025-11-28
>
> Dear Reviewer EgK3,
>
> Thank you for your constructive feedback. We also appreciate your initial assessment, particularly your comments highlighting "This method **provides a novel mechanism** to balance accuracy and computational efficiency", "... achieving **notable gains in accuracy, efficiency, and reduced latency**...", and "This **innovative approach** improves interpretability and realism of model reasoning".
>
> We would like to take this opportunity to reiterate the core contributions of our work, which we believe make a significant contribution to the field:
> * **Cognitive-inspired elastic reasoning framework.** We propose **CogER**, a four-level cognitive hierarchy that dynamically selects among reasoning modes at test time, enabling flexible control of reasoning depth and computational cost.
> * **Hierarchical reward for accuracy–efficiency trade-offs.** We design a composite reward that combines format correctness, answer accuracy, and a hierarchy-aware term based on the minimal sufficient level, explicitly discouraging overthinking while maintaining reliability.
> * **Tool-augmented deep reasoning with CoTool.** We introduce **CoTool** and the accompanying RSTKit toolkit to integrate external tools into the highest reasoning level, allowing the model to decide when to invoke tools and to incorporate tool outputs into its reasoning process.
> * **Extensive empirical validation.** Experiments on ID and OOD reasoning benchmarks show that CogER achieves stronger accuracy–efficiency trade-offs than fixed-depth and routing-based baselines.
>
> We hope that these innovations and the substantial empirical gains will merit a score improvement. Should you have any further questions, we are happy to provide additional clarifications.
>
> Thank you again for your time and consideration.
>
> Sincerely,
>
> The Authors

---

### Official Review · Reviewer_opSX · 2025-10-31

**Soundness:** 2
**Presentation:** 3
**Contribution:** 3
**Rating:** 4
**Confidence:** 3

**Summary:**

This paper proposes Cognitive-Inspired Elastic Reasoning (CogER), a framework for dynamically allocating computational resources in LLM reasoning based on query complexity. The authors classify queries into four complexity levels (L1-L4) inspired by Bloom's Taxonomy, train a 7B CogER-Agent using reinforcement learning to route queries to appropriate processing modes (ranging from direct answering to tool-augmented reasoning), and introduce Cognitive Tool-Assisted Reasoning (CoTool) for L4 queries. Experiments on mathematical reasoning and commonsense QA tasks show improvements over test-time scaling baselines with reduced computational cost.

**Strengths:**

1. **Well-Motivated Problem**: Dynamic resource allocation in test-time compute is an important practical challenge. The paper clearly articulates the inefficiency of one-size-fits-all reasoning strategies.

2. **Comprehensive System Design**: The framework includes multiple well-integrated components: complexity classification, MDP formulation, specialized reward functions (particularly R_hierarchy for cost-awareness), and CoTool for tool integration. Algorithm 1 provides clear implementation guidance.

3. **Solid Experimental Results**: Achieves competitive accuracy (89.28% on ID tasks) with significant efficiency gains (118.53s average latency vs. 506.19s for DeepSeek-R1). The OOD evaluation on MAWPS and CollegeMath demonstrates some generalization capability.

4. **Detailed Reproducibility Information**: Appendix B provides complete prompts, Appendix D specifies RSTKit tools, and implementation details (hyperparameters, training procedures) are thoroughly documented.

5. **Ablation Studies**: Tables 2-5 examine the contribution of different components (routing levels, reward terms, CoTool), providing insights into what drives performance.

**Weaknesses:**

## 1. Inadequate Baseline Selection

**Missing critical routing baselines**: The paper omits comparisons to directly relevant work:
- **RouteLLM (ICLR 2025)**: Uses preference-based training for LLM routing with similar objectives

**Unfair comparisons**:
- DeepSeek-R1 is a closed-source 671B model tested under unknown conditions; should compare against open DeepSeek-R1-Distill (7B/14B/32B)
- No iso-compute baseline: should compare "always QwQ-32B" with same average compute budget as CogER

## 2. Narrow Dataset Selection

**Limited task coverage**: Evaluation restricted to math reasoning and QA. Missing:
- Code generation (HumanEval, MBPP)
- Long-context reasoning (QuALITY, NarrativeQA)
- Multi-turn dialogue (MT-Bench)
- Factual QA with retrieval (Natural Questions, TriviaQA)
- Multimodal tasks (ScienceQA, MMMU)

**Train-test contamination risk**: Training uses samples from GSM8K/MATH/CommonsenseQA, then evaluates on same benchmark test sets. Router may learn dataset artifacts rather than complexity.

**Weak OOD evaluation**: Only 2 OOD datasets (MAWPS, CollegeMath), both still mathematical. Needs different task types.

**Small scale**: Only 8K training samples total; no ablation showing this suffices.

## 3. The L_min Determination Black Box

**Core methodology undefined**: Equation (6) depends on L_min(S) but never explains how these labels are obtained for 8,000 training samples. Possible methods, both problematic:

- **Human annotation**: No guidelines, inter-annotator agreement, or subjectivity handling reported

- **Empirical testing**: How is stochasticity handled? (Same query may succeed/fail across runs). What success threshold? (50%? 80%? 100%?). Circular dependency: need models to determine L_min for training those models.

## 4. Router Capability Paradox

**Weak-model-judging-strong-model**: 7B router must assess whether queries require 32B models or tools—but how can it judge capabilities beyond its own? Missing:
- Analysis of how 7B learns 32B/QwQ capability boundaries from binary success signals alone
- Routing accuracy stratified by true complexity (likely degrades for hard queries where accurate routing matters most)
- Router size ablation (1.8B/7B/14B/32B)

**Table 4 evidence**: Without R_hierarchy, 88.46% of queries route to L4. This reveals risk-aversion ("when uncertain, pick strongest") rather than genuine complexity understanding. R_hierarchy forces cheaper selections via penalty, not learned understanding.

**Training signal insufficiency**: GRPO provides only correct/incorrect feedback, never showing how stronger models reason. Compare to RouteLLM's preference-based training that explicitly teaches capability differences.

**No routing accuracy validation**: Table 4 shows level distributions but never reports:
- Confusion matrix vs. ground truth L_min
- Per-level precision/recall
- Error decomposition (routing mistakes vs. execution failures)

This is the fundamental metric for any routing system—its absence is critical.

**Questions:**

## Q1: L_min Label Generation Methodology

 Please provide a complete, step-by-step specification of how L_min labels are generated for the 8,000 training samples:

- What is the exact procedure? (Human annotation / Empirical testing / Heuristic rules?)

L_min is the core supervision signal and the method cannot be reproduced without this information.

## Q2: Routing Accuracy Validation

Table 4 reports level assignment distributions (2% / 28.17% / 21.90% / 47.93%) but never validates routing correctness. Please provide:

- **Confusion matrix**: Predicted level vs. ground truth L_min for test set
- **Per-level metrics**: Precision, recall, and F1 for L1/L2/L3/L4 classification

Without these metrics, we cannot assess whether the system truly learns complexity classification or succeeds through other factors (ensemble effects, tool usage).

## Q3: Router Capability Boundary Learning

The 7B router must predict whether queries require 32B/QwQ models, yet only receives binary success signals. Please address:

- **Learning mechanism**: How does the 7B router learn the capability boundaries of stronger models it never observes reasoning? GRPO provides only correct/incorrect feedback—no reasoning traces, no intermediate steps.

- **Table 4 explanation**: Without R_hierarchy, 88.46% of queries route to L4. Does this indicate:
  - The router genuinely assesses most queries as requiring tools?
  - Or risk-aversion ("when uncertain, pick the strongest option")?

  If the latter, how can we trust that R_hierarchy teaches genuine complexity understanding rather than merely forcing cheaper selections through penalty?

- **Router size ablation**: Have you tested routing with different model sizes (1.8B / 7B / 14B / 32B)?
  - If larger routers improve accuracy → confirms stronger models make better judges (but undermines cost savings)
  - If accuracy plateaus at 7B → important empirical finding worth reporting
  - If 1.8B performs comparably → suggests learned heuristics rather than deep understanding

- **Comparison to RouteLLM**: RouteLLM uses preference data to teach routers about capability differences. Why is GRPO's binary feedback sufficient when RouteLLM requires richer training signals?

## Q4: Baseline Comparisons and Fairness

- **Missing routing baselines**: Why not compare to:
  - RouteLLM (ICLR 2025) - directly comparable routing framework

- Can you compare against DeepSeek-R1-Distill models (7B/14B/32B) for fairer scale comparison?

## Q5: Generalization Beyond Training Distribution

- **Out-of-distribution tasks**: Training uses only math + commonsense + medical QA. How does CogER perform on:
  - Code generation (HumanEval, MBPP)
  - Long-context reasoning (QuALITY, NarrativeQA)
  - Multi-turn dialogue (MT-Bench)
  - Factual QA with retrieval (Natural Questions)

  The current 2 OOD datasets (MAWPS, CollegeMath) are still mathematical—not true distribution shift.

- **Training data scale**: You use 8K samples. Have you:
  - Ablated training data size to show performance saturates at 8K?
  - Tested whether more diverse training data improves routing generalization?

- **Dataset artifact learning**: Training on GSM8K/MATH then testing on same benchmarks risks learning dataset-specific patterns (e.g., "MATH → L3" heuristic). How do you ensure the router learns generalizable complexity assessment rather than dataset shortcuts?

## Q6: Model Upgrade and Long-term Viability

The 7B router is trained on specific model capabilities (Qwen2.5-7B/32B, QwQ-32B):

- **Model upgrades**: If Qwen3.0-7B surpasses Qwen2.5-32B in capability, how should the system adapt?
  - Does the router require complete retraining?
  - Can L_min labels transfer across model versions?

- **Cross-model generalization**: Can the trained router generalize to different model families (e.g., Llama, Mistral)?

- **Failure modes**: When does CogER perform worse than baselines? Please provide:
  - Case studies of routing failures
  - Characterization of query types most prone to misclassification

---

> ### Author Response · Authors · 2025-11-22
> **Response to Reviewer opSX (Part 1)**
>
> We are deeply grateful to you for recognizing the strengths of our work, particularly the **well-motivated problem** of dynamic test-time resource allocation, the **comprehensive and well-integrated system design**, the **solid experimental results**, the **detailed reproducibility information**, and the ablation studies that help clarify the contribution of each component.
>
> >Q1. The paper omits comparisons to directly relevant work: RouteLLM (ICLR 2025): Uses preference-based training for LLM routing with similar objectives.
>
> **A1.** We thank the reviewer for highlighting the importance of comparing with RouteLLM, which is indeed a closely related routing-based approach. In the revised version, we include RouteLLM as an additional baseline and report the results in Table A (also added to Table 1 in the paper). From Table A, **CogER achieves better accuracy on both ID and OOD benchmarks.**  Specifically, CogER outperforms RouteLLM on all ID datasets and on the averaged EM score (89.28 vs. 86.58 EM), and also achieves higher average performance on OD datasets (93.56 vs. 92.60 EM). This indicates that even when compared against a strong routing-based method with similar efficiency objectives, CogER provides a better accuracy.
>
> Table A: Comparison between CogER and the LLM routing baseline RouteLLM on ID and OOD benchmarks.
> | Baseline           | GSM8K        | MATH         | Com-QA        | MedQA        | AVG (ID)      | MAWPS        | College      | AVG (OOD)    |
> |--------------------|-------------:|-------------:|--------------:|-------------:|--------------:|-------------:|-------------:|-------------:|
> | RouteLLM           | 95.80±0.05   | 87.29±0.12   | 83.99±0.19    | 79.22±0.32   | 86.58±0.20    | **97.90±0.00**   | 87.29±0.08   | 92.60±0.06   |
> | **CogER (Ours)**   | **96.18±0.05** | **95.20±0.20** | **84.52±0.30** | **81.23±0.00** | **89.28±0.18** | 97.87±0.01 | **89.24±0.14** | **93.56±0.10** |
>
>
> >Q2. Unfair comparisons:1) DeepSeek-R1 is a closed-source 671B model tested under unknown conditions; should compare against open DeepSeek-R1-Distill (7B/14B/32B). 2) No iso-compute baseline: should compare "always QwQ-32B" with same average compute budget as CogER.
>
> **A2.** We thank the reviewer for raising these important concerns about the fairness of our comparisons. We clarify the role of DeepSeek-R1, our use of DeepSeek-R1-Distill baselines, and the iso-compute comparison with “always QwQ-32B” as follows.
> * **Use of DeepSeek-R1 as a strong reference system.** We include DeepSeek-R1 (671B) because it is one of the most powerful publicly reported reasoning-centric LLMs and therefore serves as a meaningful reference point. From Table 1, **compared to DeepSeek-R1, CogER achieves a relative performance improvement of 9.48% in terms of average EM on ID tasks**, while using substantially lower average compute.
> * **Comparisons with open DeepSeek-R1-Distill models.** To provide fully reproducible and model-family-consistent baselines, we also compare against DeepSeek-R1-Distill 7B, 14B, and 32B. These results are reported in **Table 1** in the paper. Our **CogER consistently outperforms the DeepSeek-R1-Distill models** in terms of average EM.
> * **Iso-compute comparison with the “always QwQ-32B” baseline.** We agree that iso-compute baselines are important for a fair assessment. To address this, we explicitly include an **“always QwQ-32B”** baseline in **Table 2**, which uses the same QwQ-32B model as our higher levels and is evaluated under a comparable average compute budget. From Table 2, under a similar average compute in terms of latency and token usage, CogER achieves a higher average EM than “always QwQ-32B”.

---

> ### Author Response · Authors · 2025-11-22
> **Response to Reviewer opSX (Part 2)**
>
> >Q3. Training uses samples from GSM8K/MATH/CommonsenseQA, then evaluates on same benchmark test sets. Router may learn dataset artifacts rather than complexity. Missing: 1) Code generation (HumanEval, MBPP); 2)Long-context reasoning (QuALITY, NarrativeQA); Multi-turn dialogue (MT-Bench); Factual QA with retrieval (Natural Questions, TriviaQA); Multimodal tasks (ScienceQA, MMMU).
>
> **A3.** We thank the reviewer for raising this concern about overfitting to specific benchmarks and for suggesting a broader evaluation. We respond from three aspects and have added the corresponding results in the revised version (**Sec. 5.3**).
> * **Evidence that the CogER goes beyond dataset-specific artifacts.** In the main paper, CogER is trained on GSM8K, MATH, Com-QA, and MedQA, but is evaluated not only on their test sets, but also on out-of-domain math datasets such as MAWPS and College. CogER consistently improves both ID and OOD tasks under a reduced compute budget, which suggests that the learned level-selection policy generalizes across related but distribution-shifted tasks rather than memorizing idiosyncratic patterns of a single benchmark.
> * **New experiments on code, long-context, and retrieval-augmented QA.** We further evaluate CogER on three additional benchmarks that were not used for training, covering code generation, long-context reasoning, and retrieval-augmented factual QA. From Table B, CogER matches or surpasses strong baselines on MBPP (code generation, Pass@3), QuALITY (long-context multiple-choice QA), and Natural Questions (retrieval-based QA). These results indicate that the cognitive hierarchy and elastic level selection transfer to tasks with very different formats and reasoning requirements, rather than being tied to the GSM8K/MATH/Com-QA domains.
> * **Scope and limitations regarding multi-turn dialogue and multimodal tasks.** We agree that multi-turn dialogue and multimodal reasoning are highly relevant and valuable directions. In this work, however, we intentionally scope CogER to single-turn, text-only settings to cleanly study cognitive-inspired elastic reasoning for LLMs under controlled conditions and to avoid confounding factors from dialogue state tracking or visual perception. This design choice and its limitations are explicitly discussed in our “Discussions and Future Works” section, where we outline extending CogER to interactive and multi-modal settings as an important avenue for future research.
>
> Table B:  Performance of CogER and baseline models on additional benchmarks, including code generation (MBPP, Pass@3), long-context reasoning (QuALITY, accuracy), and retrieval-augmented factual QA (Natural Questions, F1-score).
>
> |Baseline|MBPP (pass@3)|QuALITY (ACC)|Natural Questions (F1)|
> |-|-|-|-|
> |Math-72B|75.16|55.77|45.52|
> |DS-R1-DQ-7B|77.30|35.21|15.41|
> |DS-R1-DQ-14B|79.88|73.33|61.14|
> |DS-R1-DQ-32B|91.22|81.22|64.45|
> |L1-MAX|38.30|25.45|2.96|
> |S1-32B|86.00|81.41|66.99|
> |ReasonFlux-32B|91.70|81.88|63.37|
> |RouteLLM|90.68|**82.97**|67.06|
> |CogER (Ours)|**91.76**|**82.97**|**67.25**|
>
>
> >Q4. Only 8K training samples total; no ablation showing this suffices.
>
> **A4.** We thank the reviewer for raising the question about whether 8K training samples are sufficient. We added a data-scale ablation to make this explicit (**see Appendix F**).
> * **Ablation over training set size.** We vary the number of training samples while keeping the data mixture and all other settings fixed. As shown in Table C, CogER already achieves strong performance with as few as 4K samples, and that performance is relatively stable across different data scales. The 8K configuration attains the best accuracy on both ID and OOD benchmarks, which supports our choice in the main paper.
> * **Sample efficiency and supervision signals.** Each training instance provides rich supervision for the policy: a binary correctness signal from the final answer and the hierarchical preference encoded by $L_{\min}(S)$ in $\mathcal{R}_{\text{hierarchy}}$. The policy only needs to choose among four fixed reasoning modes rather than learn the underlying tasks from scratch, so the sample complexity is modest. As shown in Table C, performance already plateaus around 8K examples, and the small fluctuations at 10K are within the variance one would expect from our limited training budget and random initialization.
>
> Table C: Effect of training set size on performance of CogER (EM, %).
> |Dataset size|4K|6K|8K|10K|
> |-|-|-|-|-|
> |ID|86.48|86.51|**89.28**|86.48|
> |OOD|92.73|92.75|**93.56**|92.69|

---

> ### Author Response · Authors · 2025-11-22
> **Response to Reviewer opSX (Part 3)**
>
> >Q5. Equation (6) depends on L_min(S) but never explains how these labels are obtained for 8,000 training samples. Possible methods, both problematic: **Human annotation:** No guidelines, inter-annotator agreement, or subjectivity handling reported. **Empirical testing**: How is stochasticity handled? (Same query may succeed/fail across runs). What success threshold? (50%? 80%? 100%?). Circular dependency: need models to determine L_min for training those models.
>
> **A5.** We thank the reviewer for carefully examining the role of $L_{\min}(S)$ in Equation (6). We clarify how $L_{\min}(S)$ is constructed and used in our framework from three aspects.
> * **Construction of $L_{\min}(S)$ via empirical testing.**  We do not rely on human annotation to obtain $L_{\min}(S)$. Instead, $L_{\min}(S)$ is computed before RL training by running each training query $S$ under the four fixed levels $L_1–L_4$ with the corresponding reasoning strategies and underlying models. Concretely, for every training sample $S$, we evaluate the base system once at each level $L_1, L_2, L_3, L_4$ and check whether the final answer is correct using the exact match metric. We then define $L_{\min}(S)$ as **the smallest level among $\{L_1, L_2, L_3, L_4\}$ that yields a correct answer under this fixed configuration**.
> * **Handling of stochasticity and the success criterion.** All evaluations used to derive $L_{\min}(S)$ are performed with **deterministic decoding settings** (e.g., **temperature = 0**, which removes sampling randomness, and a fixed decoding configuration). Under these settings, the same query at the same level produces the same answer, so there is no run-to-run randomness when computing $L_{\min}(S)$. The success criterion is therefore straightforward: a level is considered successful for a query $S$ if, under this deterministic configuration, the level produces an answer whose exact match equals 1. We do not rely on multiple stochastic runs and thus do not need to define a 50% or 80% success threshold.
> * **Avoiding circular dependency when using $L_{\min}(S)$.** During RL training, the parameters of the underlying reasoning models for L2–L4 remain fixed, and only the routing policy of the CogER-Agent is updated. Thus, $L_{\min}(S)$ is derived from the fixed capabilities of the base models and is used as a **shaping signal** in the hierarchical reward to encourage the agent to choose the lowest sufficient level, without changing how accuracy is evaluated.
>
> >Q7. Routing accuracy stratified by true complexity (likely degrades for hard queries where accurate routing matters most).
>
> **A7.** We thank the reviewer for this helpful suggestion. Our clarification is as follows:
>
> * **Accuracy for queries routed to each level.** For every test query, we record the level chosen by the CogER-Agent and then compute the exact-match accuracy within the subset of queries routed to that level. Denoting these quantities as $\text{Acc}(L_i)$, Table D reports the results on both ID and OOD benchmarks. We observe that the accuracy within each routed subset is consistently high. These results indicate that, conditioned on the router’s decision, **each reasoning mode is applied to a subset of queries on which it is mostly successful.**
> * **Role of $L_{\min}(S)$ and routing objective.** In our framework $L_{\min}(S)$ is used in the Hierarchical-Aware Reward $R_{\text{hierarchy}}$ (Eq. (6)–(8)) as a preference signal that **encourages the agent to avoid unnecessarily high levels when they are not needed**. The policy is still optimized with respect to the combined reward $R_{\text{format}} +R_{\text{accuracy}} + R_{\text{hierarchy}}$. In other words, the **training objective is to maximize answer accuracy while reducing computational cost**, rather than to exactly classify each query to its estimated $L_{\min}(S)$. Therefore, we do not expect the router to perfectly match $L_{\min}(S)$ on every example, and slight deviations toward stronger levels are acceptable when they improve reliability. Under this setting, the strict top-1 routing accuracy with respect to $L_{\min}(S)$ is 38.88%, which is expected because the policy is not trained as a pure classifier of $L_{\min}(S)$, while the **task-sufficient routing accuracy**, defined as routing to a level that successfully solves the query, **reaches 93.93%**.
>
> Table D:  Accuracy (%) of queries routed by the CogER-Agent to each reasoning level on ID and OOD benchmarks. $\text{Acc}(L_i)$ denotes the exact-match accuracy computed over the subset of queries that are dynamically routed to level $L_i$.
>
> |Version|ID|OOD|
> |-|-|-|
> |$\text{Acc}(L_1)$|95.45|100.00|
> |$\text{Acc}(L_2)$|92.45|97.20|
> |$\text{Acc}(L_3)$|95.82|95.06|
> |$\text{Acc}(L_4)$|92.60|90.64|

---

> ### Author Response · Authors · 2025-11-22
> **Response to Reviewer opSX (Part 4)**
>
> >Q6. How does the 7B router learn the capability boundaries of stronger models it never observes reasoning? GRPO provides only correct/incorrect feedback—no reasoning traces, no intermediate steps.
>
> **A6.** We thank the reviewer for this thought-provoking question. Our 7B CogER-Agent does not attempt to explicitly model the internal reasoning process of stronger models. Instead, it learns a routing policy that maps queries to levels based on outcome driven feedback. We explain this from three aspects.
> * **Learned routing policy rather than explicit internal capability modeling.** As described in Sec. 4.2, the CogER-Agent is trained to choose among the four reasoning levels $L_1$ to $L_4$ corresponding to No-Think, Think, Extend, and Delegate for each query, by producing the appropriate `<question_level>Li</question_level>` tag in the system prompt. The objective of the router is therefore to learn a query to level decision boundary. It needs to learn for which types of queries a low cost level is sufficient and for which types a higher level is necessary to obtain correct answers. In this sense, the capability boundaries of stronger models are captured implicitly at the behavioral level. Different levels succeed or fail on different queries, rather than being modeled through explicit reconstruction of internal reasoning traces.
> * **Information provided by correct or incorrect feedback and hierarchy-aware reward.** Although GRPO uses final correctness as the primary task signal, the composite reward in Eq. (6) is richer than a single zero one label. It combines an accuracy term that indicates whether the chosen level solves the query and a hierarchy-aware term that encourages the agent to use the lowest sufficient level, based on $L_{\min}(S)$, and penalizes overthinking. During training, queries from the same distribution are encountered repeatedly. If the router often selects a low level for a certain pattern of queries and receives incorrect rewards, GRPO will push the policy toward higher levels for that pattern. Conversely, if a higher level is frequently chosen when a lower level already suffices, the hierarchy-aware reward reduces the return of that choice and moves the policy toward cheaper levels.  From Tables 2&4, the learned policy is not degenerate. It neither always chooses the highest level nor always chooses $L_1$. This supports that these outcome based signals are sufficient for the router to approximate effective capability boundaries.
> * **Lack of reasoning traces as a deliberate modular design choice.** The router does not observe the full chain of thought or intermediate steps of the stronger models at $L_3$ and $L_4$. This is a deliberate design choice. CogER is intended to treat the underlying reasoning modules as black box experts and to remain agnostic to their internal formats. The router only needs to observe the input query and the final outcome, that is, the reward, for each level choice, rather than the internal reasoning paths. This keeps the framework modular. One can replace the back-end CoT models or tool-augmented modules with stronger experts and simply rerun rollout and RL with the new experts, without changing the router architecture.
>
>
> >Q8. Router size ablation.
>
> **A8.** We thank the reviewer for asking about the impact of router size. We have added a router size ablation and summarize our findings as follows.
> * **Effect of router size on performance.** We train CogER with three router sizes while keeping all other components fixed. As shown in Table E, the MAWPS accuracy is 97.06% with a 3B router, 97.87% with the 7B router used in the main paper, and 98.32% with a 14B router. This shows that CogER is **robust across router sizes** and that the performance gains from enlarging the router are modest rather than dramatic.
> * **Choice of 7B as a practical trade-off.** The above trend indicates **diminishing returns** when scaling the router. Moving from 3B to 7B brings a small improvement, while further scaling to 14B yields only a marginal gain at additional compute cost. We therefore keep the 7B router in the main experiments as a balanced choice that already achieves strong performance without making the routing module disproportionately large relative to the experts.
>
> Table E: Effect of router size on MAWPS accuracy of CogER.
> |Router size|MAWPS|
> |-|-|
> |3B|97.06|
> |7B（in paper）|97.87|
> |14B|98.32|

---

> ### Author Response · Authors · 2025-11-22
> **Response to Reviewer opSX (Part 5)**
>
> >Q9. Table 4 evidence: Without R_hierarchy, 88.46% of queries route to L4. This reveals risk-aversion ("when uncertain, pick strongest") rather than genuine complexity understanding. R_hierarchy forces cheaper selections via penalty, not learned understanding.
>
> **A9.** We thank the reviewer for this insightful comment, which directly relates to our core goal of achieving cognitively inspired elastic reasoning rather than a one-size-fits-all strategy.
> * **Table 4 highlights the overthinking problem that CogER is designed to solve.** Without $R_{\text{hierarchy}}$, the policy indeed routes most queries to $L_4$. This behavior reflects the overthinking phenomenon that motivates our work, where optimizing purely for accuracy naturally drives the system to always choose the strongest and most expensive reasoning mode. Therefore, Table 4 supports our problem formulation and shows why an additional hierarchy-aware signal is necessary to realize elastic reasoning across the four levels.
> * **The hierarchy reward guides but does not hard-code complexity-aware routing.** $R_{\text{hierarchy}}$ does not fix the level for any query. It is a soft cost term that trades off with the accuracy reward. If the router sends a query to a cheaper level that fails, the drop in accuracy makes the total reward low. To obtain a high return, the agent still has to learn which queries can be reliably solved at lower levels and when it is safer to stay at $L_4$. With $R_{\text{hierarchy}}$, CogER uses all four levels and at the same time keeps or improves accuracy compared with the variant that almost always chooses $L_4$. This pattern indicates that the agent learns a non-trivial routing policy instead of only avoiding penalties.
>
>
> >Q10. RouteLLM uses preference data to teach routers about capability differences. Why is GRPO's binary feedback sufficient when RouteLLM requires richer training signals?
>
> **A10.** We thank the reviewer for raising this insightful comparison with RouteLLM. We clarify why GRPO with our composite reward is sufficient in our setting and how this differs from the preference based training used by RouteLLM, as follows.
> * **Composite reward and GRPO optimization.** The CogER Agent is trained with the combined reward $R_{\text{format}} +R_{\text{accuracy}} + R_{\text{hierarchy}}$ under GRPO rather than a single binary signal. For each query, we generate multiple rollouts that use different reasoning levels and compute rewards within the same GRPO group. Trajectories that solve the query at a lower level obtain a higher total return because they receive a positive accuracy reward and a smaller hierarchical penalty. In contrast, trajectories that fail or rely on unnecessarily high levels receive a lower return. This group relative optimization induces a preference ordering among levels for each query, similar in spirit to preference data without explicitly constructing pairwise preference labels.
> * **Structured four-level hierarchy simplifies routing supervision.** RouteLLM learns routers from human preference data to distinguish capability differences between candidate models. In our CogER, the four routing options are not arbitrary experts but cognitively motivated reasoning modes L1–L4 that are defined by their behavior, including prompt answering, concise chain of thought, deep chain of thought, and tool-enhanced reasoning, with a monotonic increase in compute and typical capability. The estimated minimal sufficient level $L_{\min}(S)$ and the hierarchical aware reward further encode the preference to use the simplest level that is usually sufficient for each query. This structured hierarchy and reward design provide a strong inductive bias for routing and make it possible to train an effective CogER Agent with GRPO without requiring additional human preference labels. At the same time, the concrete base models used to implement each level can be replaced or upgraded without changing this routing principle.
> * **Empirical evidence that GRPO-based feedback is effective.** To directly address the concern, we include RouteLLM as an additional routing baseline in the revised version (Table 1 in the paper). From Table A, under the same backbone and evaluation setup, **CogER outperforms RouteLLM** on both ID and OOD benchmarks. These results indicate that in our cognitive hierarchy setting, the GRPO-based correctness signal combined with the hierarchical aware reward is sufficient to learn an effective router in practice, even though RouteLLM uses richer human preference supervision.

---

> ### Author Response · Authors · 2025-11-22
> **Response to Reviewer opSX (Part 6)**
>
> >Q11. Table 4 shows level distributions but never reports: 1）Confusion matrix vs. ground truth L_min. 2) Per-level precision/recall. 3) Error decomposition (routing mistakes vs. execution failures). This is the fundamental metric for any routing system—its absence is critical.
>
> **A11.**  We thank the reviewer for emphasizing that a detailed analysis of routing quality is fundamental.
> * **Confusion matrix and per-level precision recall.** In Appendix F, we additionally provide the full 4×4 confusion matrix between the CogER-Agent’s predicted levels. Table F summarizes per-level precision, recall, and F1 score. We observe that $L_1$ and $L_2$ achieve very high precision (97.53% and 96.81%), which means that when the agent decides a query can be solved at a lower level, this prediction is usually correct. Levels $L_3$ and $L_4$ exhibit higher recall, which shows that most oracle hard queries are indeed routed to higher levels, while their lower precision mainly reflects our cost-aware training objective that allows conservative routing to stronger levels when necessary.
> * **Consistency with the routing objective of CogER.** As discussed in Sec. 4.3, $L_{\min}(S)$ is used inside the Hierarchical Aware Reward as a preference signal that encourages the agent to avoid unnecessarily high levels while maintaining accuracy. The policy is optimized for the combined reward $R_{\text{format}} +R_{\text{accuracy}} + R_{\text{hierarchy}}$, so the primary objective is to maximize answer accuracy under limited compute rather than to exactly classify every query to its oracle $L_{\min}(S)$. The new diagnostics confirm that the router behaves in a reasonable and interpretable way under this objective, while the overall system still achieves state-of-the-art accuracy with significantly reduced compute.
> * **Quantitative breakdown of failure sources.**  CogER achieves 89.28 EM on ID tasks and 93.56 EM on OOD tasks. Among the remaining errors, on ID tasks, 51.38% of failures are due to incorrect routing (the agent selects an insufficient level), and 48.62% are due to execution errors given a correct or more conservative level. On OOD tasks, 48.18% of failures are attributed to routing and 51.82% to execution. This analysis shows that both components contribute to the residual error.  We will report these numbers in the revised version (Appendix F) to make the behavior of CogER more transparent.
>
> Table F: Per level routing precision(%), recall(%), and F1 score(%) of the CogER-Agent on all benchmarks.
> |Version|Precision|Recall|F1-score|
> |-|-|-|-|
> |$L_1$|97.53|39.27|55.99|
> |$L_2$|96.81|98.68|97.73|
> |$L_3$|20.80|85.09|33.43|
> |$L_4$|36.67|4.59|8.15|

---

> ### Author Response · Authors · 2025-11-22
> **Response to Reviewer opSX (Part 7)**
>
> >Q12. Model upgrades: If Qwen3.0-7B surpasses Qwen2.5-32B in capability, how should the system adapt? Does the router require complete retraining? Can L_min labels transfer across model versions?  Can the trained router generalize to different model families (e.g., Llama, Mistral)?
>
> **A12** We thank the reviewer for raising the important question of how CogER behaves under model upgrades. We clarify three aspects: how the hierarchy is defined, how CogER adapts to upgraded backbones in practice, and how $L_{\min}$ labels can be reused.
> * **Hierarchy defined by reasoning modes rather than fixed backbones.** In our framework, the four levels $L_1–L_4$ are defined as **reasoning modes** (prompt answering, concise CoT, deep CoT, and tool-enhanced reasoning, see Sec. 4.1), not as specific LLMs. The policy is trained to choose among these modes using the combined reward $R_{\text{format}} +R_{\text{accuracy}} + R_{\text{hierarchy}}$. As long as each level continues to implement the same mode, the underlying model at that level can be upgraded or replaced without changing the overall architecture of CogER.
> * **Adapting to model upgrades in practice.** To directly test how well CogER tolerates backbone changes **without retraining the CogER-Agent**, we replace the models used at different levels and keep the learned policy fixed. In particular, we plug a LLaMA 3.1 70B model into the $L_1$ slot and a Qwen3-8B model into the $L_2$ slot, while retaining QWQ-32B and our CoTool for $L_3$ and $L_4)$. As shown in Table G, CogER still achieves 90.02% EM on CollegeMath under this new configuration, indicating that the learned policy can be directly reused after reasonable model upgrades without retraining while maintaining strong performance.
> * **Reusing and adapting $L_{\min}$ labels.** The labels $L_{\min}(S)$ are estimated with respect to a given model stack and are used in $R_{\text{hierarchy}}$ as a **preference signal** that encourages avoiding unnecessarily high levels while maintaining accuracy. When the backbone models are upgraded, the original $L_{\min}(S)$ remain **conservative upper bounds** on the difficulty of each query. They can therefore be reused as an initialization of the reward design, and if necessary refined on a small set of examples with the new models. This means that model upgrades do not require recomputing all labels from scratch and do not force a full retraining of the router.
>
>
> Table G: Performance of CogER after upgrading the level backbones without retraining the CogER-Agent.
> |Version|CollegeMath|
> |-|-|
> |$L_1$(LlaMA 3.1 70b)|72.67|
> |$L_2$(Qwen3-8b)|86.50|
> |$L_3$(QWQ-32B)|87.17|
> |$L_4$(Our CoTool)|89.75|
> |**CogER**|**90.02**|

---

> ### Author Response · Authors · 2025-11-22
> **Response to Reviewer opSX (Part 8)**
>
> >Q13. When does CogER perform worse than baselines? Please provide: 1) Case studies of routing failures. 2) Characterization of query types most prone to misclassification.
>
> **A13.** We thank the reviewer for asking when CogER fails and for encouraging a more concrete analysis of its error modes. In the revision, we add a qualitative study of typical failures and summarize the query types that are most prone to misclassification.
> * **Case studies of where CogER fails.** We include two representative failure cases in the appendix:
>     * **CommonsenseQA: inherent reasoning limitation.** In this example, the question requires nuanced commonsense reasoning about the *main* benefit of exercising. All four levels $L_1–L_4$ (from direct answering to tool-enhanced reasoning) produce the same incorrect option, because the underlying LLM consistently prefers a plausible but wrong explanation. In other words, there is **no valid level among $\{L_1,\dots,L_4\}$** that can solve this query; any routing decision is doomed to fail. This illustrates a regime where CogER can underperform very strong baselines simply because of the intrinsic limitations of the base experts, rather than routing failures.
>     * **MAWPS: routing error leading to faulty reasoning.** The second example is an arithmetic word problem with distractor information (“6 were torn”). Ideally, it should be handled by a higher level (e.g., $L_3$ or $L_4$) that performs more careful multi-step reasoning or uses tools. However, the policy routes it to a lower level, which produces a shallow chain of thought that ignores the subtraction step and yields an incorrect answer. This case highlights a **genuine routing failure**, where the chosen level is not powerful enough, even though a higher level in CogER could have solved the problem.
> * **Query types most prone to misclassification.** From these and other inspected cases, we observe that misrouted queries typically fall into two broad categories:
>     * **Borderline-complex questions near level boundaries.** Problems whose difficulty lies between two neighboring levels (e.g., “L2 vs. L3”) are most likely to be misclassified—especially arithmetic or multi-step questions with subtle distractors, where it is hard to decide in advance whether concise or deep reasoning is needed.
>     * **Conceptually tricky commonsense or knowledge-intensive questions.** For some CommonsenseQA-style items, the true difficulty comes from missing or implicit world knowledge rather than the formal number of reasoning steps. In such cases, even routing to the highest level $L_4$ may not help, because all experts share the same knowledge gap. These queries contribute to the portion of errors that we attribute to the execution modules rather than to routing.
>
> Overall, this analysis shows that CogER’s mistakes are concentrated on **borderline and knowledge-hard queries**, instead of arising from a degenerate “always pick the cheapest / always pick the strongest” behavior. We will incorporate these case studies and the above characterization into the appendix to make the limitations of CogER more explicit and to motivate future work on stronger experts and more fine-grained level definitions.
>
> ***
> We sincerely hope our clarifications above have addressed your concerns. We would be grateful if you could kindly reconsider the evaluation of our paper.

---

> ### Author Response · Authors · 2025-11-25
> **Looking forward to the response from Reviewer opSX**
>
> Dear Reviewer opSX,
>
> We have addressed your initial concerns regarding our paper. We are happy to discuss them with you in the OpenReview system if you feel that there are still some concerns/questions. We also welcome new suggestions/comments from you!
>
> Best regards,
>
> The Authors

---

> ### Author Response · Authors · 2025-11-28
>
> Dear Reviewer opSX,
>
> Thank you for your constructive feedback. We also appreciate your initial assessment, particularly your comments highlighting "**Well-Motivated Problem**", "**Comprehensive System Design**", "**Solid Experimental Results**", and "**Detailed Reproducibility Information**".
>
> In the rebuttal, we have provided detailed, point-by-point responses to all your comments and made revisions to the manuscript, mainly including:
> * **Stronger and fairer baseline.** We add RouteLLM as a directly comparable routing baseline and report results on both ID and OOD datasets. CogER achieves higher EM than RouteLLM on both ID and OOD benchmarks.
> * **Broader evaluation beyond training benchmarks.** We evaluate CogER on additional benchmarks, including MBPP (code generation), QuALITY (long-context QA), and Natural Questions (retrieval-augmented QA), to demonstrate generalization beyond the training distribution.
> * **Clearer definition and use of $L_{\min}(S)$.** We explain in detail how $L_{\min}(S)$ is obtained via deterministic runs of the four fixed levels (no human annotation or stochastic thresholds) and clarify that it is only used as a shaping signal in the hierarchical reward, not as a test-time oracle.
> * **Comprehensive routing diagnostics and ablations.** We add per-level precision/recall/F1, accuracy for queries routed to each level, a decomposition of errors into routing vs. execution, data-scale and router-size ablations, and a study of robustness under model upgrades (e.g., replacing back-end experts without retraining the router).
> * **Failure case analysis.** We provide representative failure cases and characterize the types of queries most prone to misrouting (e.g., borderline-complex and knowledge-hard questions), to make the limitations of CogER more explicit.
>
> We are confident that these revisions address your concerns and greatly improve the clarity of our work.
>
> Thank you again for your time and consideration.
>
> Best regards,
>
> The Authors

---

### Author Response · Authors · 2025-11-27
**General Response**

Dear ACs and Reviewers,

We sincerely appreciate your time and effort in reviewing our paper and providing constructive feedback. Besides the response to each reviewer, here we would like to further 1) thank reviewers for their recognition of our work and 2) highlight the major modifications in our revision:

**1. We are glad that all reviewers appreciate and recognize our novelty and contribution.**
* "**Well-Motivated** Problem:...  clearly articulates the inefficiency of one-size-fits-all reasoning strategies."; "**Clear Motivation & Principled Design**: ..."[Reviewers opSX, Y9xi]
* "... provides **a novel mechanism** ..."; "This **innovative approach** improves interpretability and realism of model reasoning."; "**a novel classification perspective** and a **new method** for understanding problem difficulty" [Reviewers EgK3, Bhkw]
* "**Solid Experimental Results**: Achieves competitive accuracy ... significant **efficiency gains** ... emonstrates some **generalization capability**."; "... achieving **notable gains in accuracy, efficiency**, and **reduced latency** compared to standard fixed or scaling-based strategies."; "... **SOTA accuracy** on both ID and OOD tasks. The **efficiency gains are significant**, ..." [Reviewers opSX, EgK3, Y9xi]

**2. We summarize the main modifications in our revised paper (highlighted in blue).**
* **Stronger and fairer baseline.** We add RouteLLM as a directly comparable routing baseline and report results on both ID and OOD datasets, and we update **Table 1** accordingly to include this stronger baseline. [Reviewers opSX]
* **Broader evaluation.** We evaluate CogER on additional benchmarks, including MBPP (code generation), QuALITY (long-context QA), and Natural Questions (retrieval-augmented QA), to demonstrate generalization beyond the training distribution, and we report these results in the newly added **Table 6**. [Reviewers opSX]
* **Efficiency and scalability analyses.** We include a controlled latency breakdown showing that the CogER-Agent adds only ~0.01s overhead per query, a data-scale ablation on the number of training samples, and a router-size ablation showing modest gains beyond 7B. The corresponding results are reported in **Tables 10, 12, and 14**.[Reviewers opSX, Y9xi]
* **Clearer definition of the minimal sufficient level.** We provide a detailed description of how $L_{min}$ is estimated via deterministic runs of the four fixed levels, explain how stochasticity is avoided, and clarify that $L_min$ is only used as a shaping signal in the hierarchical reward, not as a test-time oracle. These details are provided in **Appendix D.3**.  [Reviewers opSX, EgK3, Bhkw]
* **Comprehensive routing diagnostics.** We add (i) an oracle upper bound with perfect routing, (ii) per-level precision/recall/F1, and (iii) per-level accuracy for queries routed to each level, together with a quantitative breakdown of failures into routing vs. execution errors. The corresponding analyses are reported in **Tables 10, 11, 13, and 15**. [Reviewers opSX, Y9xi]
* **Robustness, design clarifications, and limitations.** We clarify the design of CoTool and the L4 mode, describe how we handle tool errors and prevent format-reward gaming, and explain the factorized MDP action space. We further study robustness under model upgrades and add qualitative failure cases plus a more explicit discussion of current limitations and future work. These updates are reflected in **Section 4.2, Appendix D.1, Appendix E, and Appendix G.1**. [Reviewers opSX, EgK3, Bhkw]


Best regards,

The Authors

---

### Note · Program_Chairs · 2026-01-17
**Submission Desk Rejected by Program Chairs**

The following references in this submission do not refer to real documents and/or have major errors in bibliographic information:

 X. Zhang, Y. Li, and Q. Wang. Group relative policy optimization. In Proceedings of the 41st International Conference on Machine Learning, 2024.